# Neighborhood Stability as a Measure of Nearest Neighbor Searchability

## Abstract

Clustering-based Approximate Nearest Neighbor Search (Anns) organizes a set of points into partitions, and searches only a few of them to find the nearest neighbors of a query. Despite its popularity, there are virtually no analytical tools to determine the suitability of clustering-based Anns for a given dataset—what we call "searchability." To address that gap, we present two measures for flat clusterings of high-dimensional points in Euclidean space. First is Clustering-Neighborhood Stability Measure (clustering-Nsm), an internal measure of clustering quality—a function of a clustering of a dataset—that we show to be predictive of Anns accuracy. The second, Point-Neighborhood Stability Measure (point-Nsm), is a measure of clusterability—a function of the dataset itself—that is predictive of clustering-Nsm. The two together allow us to determine whether a dataset is searchable by clustering-based Anns given only the data points. Importantly, both are functions of nearest neighbor relationships between points, not distances, making them applicable to various distance functions including inner product.

## 1 Introduction

Clustering has a wide array of applications including classification and Approximate Nearest Neighbor Search (Anns). In clustering-based Anns (Vecchiato, 2024; Bruch, 2024, Chapter 7), a set of $m$ data points are first clustered into $\sqrt{m}$ clusters during the indexing phase (Douze et al., 2024, Section 5.1), and only a fraction of the resulting clusters are searched to find the $k$ nearest neighbors of a query. Search is naturally approximate and its efficacy is measured by recall or accuracy: if $S$ is the set of exact $k$ nearest neighbors and $S'$ is the set returned by Anns, then accuracy is $|S \cap S'|/k$.

Clustering-based Anns itself has received much attention in the literature (Auvolat et al., 2015; Babenko & Lempitsky, 2012; Chierichetti et al., 2007; Bruch et al., 2024b; Douze et al., 2024) and is a popular paradigm in industry.[1] Despite its wide adoption, determining whether clustering-based Anns is appropriate for a dataset, or pinpointing which clustering algorithm is the right choice, have remained empirical questions that require extensive experiments and large query sets.

We focus on this gap in our work and present measures that serve as the first tools to determine the *searchability* of a dataset by clustering-based Anns given only the dataset itself. Our measures build on the intuition that a better clustering leads to higher Anns accuracy, and that datasets that can be partitioned into higher-quality clusters enjoy higher searchability. How do we quantify these notions?

### 1.1 Clustering quality

There is a suite of measures in the literature that assess the quality of a clustering. Clustering quality measures, as they are known, fall into one of *internal* or *external* categories. External measures use task-related labels to evaluate how well a given clustering models the data. Such measures answer questions such as: Are clusters *pure* and contain only points with the same label? Does a clustering of data yield a higher Anns accuracy? That is the paradigm used by researchers and practitioners to evaluate clustering-based Anns on a dataset.

---

[1]See, for example, `https://turbopuffer.com/blog/turbopuffer`, `https://www.pinecone.io/blog/serverless-architecture/` `#Pinecone-serverless-architecture`, and `https://research.google/blog/announcing-scann-efficient-vector-similarity-search/`.

Internal quality measures, on the other hand, use no auxiliary information and as such are task-independent. They answer questions such as: Are clusters compact and well-separated? The unsupervised nature of internal measures makes them universally applicable and therefore more attractive, explaining, among other research, attempts to formalize this notion through an axiomatic approach (Ben-David & Ackerman, 2008; Kleinberg, 2002; Van Laarhoven & Marchiori, 2014).

That distinction between external and internal measures is relevant to our study. In order to assess whether a clustering has higher quality and conclude that ANNS over that clustering has higher accuracy, all without access to a query set, implies that we must look for an internal measure. As we show later, however, existing internal quality measures are not predictive of ANNS accuracy.

## 1.2 CLUSTERABILITY

We can go one step further and assess the *clusterability* or *clustering tendency* of a set of points (Adolfsson et al., 2019; Ackerman & Ben-David, 2009). Whereas clustering quality measures are functions of a particular clustering—that is, they take a clustering of a set of points and compute its quality—clusterability measures determine whether a set of points can be meaningfully clustered by any clustering algorithm. A higher clusterability implies that a clustering can emerge with an optimal loss or high clustering quality (according to some clustering quality measure).

Ideally, high (low) clusterability should be indicative of high (low) internal clustering quality of a clustering, which in turn should hint at high (low) external clustering quality of the same clustering given a task. When applied to ANNS, that logic implies a highly clusterable dataset should ideally yield high ANNS accuracy for any query distribution—rendering it, in our terminology, searchable.

That ideal, however, typically does not materialize. There is often a disconnect between clusterability measures, internal quality measures, and external ones. Furthermore, clusterability and internal quality measures often require the distance function to be nonnegative—a constraint that renders existing measures unsuitable in inner product-based ANNS.

## 1.3 OUR CONTRIBUTION

In this work, we focus exclusively on *flat* clusterings of points, and consider its application to Euclidean space with three widely-used distance and similarity functions in ANNS: Euclidean distance; cosine similarity; and, inner product. Within this context, we present an internal measure of clustering quality together with a measure of clusterability, addressing the gaps enumerated above.

Our internal measure of clustering quality is based on a modification of the notion of $k$-Nearest Neighbor ($k$-NN) *consistency* (Ding & He, 2004). A set is said to be $k$-NN consistent if for every point in the set, its $k$ nearest neighbors also belong to the set. We relax this definition by forgoing (binary) consistency and fixing $k = 1$, and instead measuring the fraction of points whose 1-NN belongs to the set. We call this set-Neighborhood Stability Measure (NSM). Our clustering quality measure, clustering-NSM, is a weighted average of the set-NSM of all clusters in a clustering.

We prove that clustering-NSM satisfies the axioms proposed by Ben-David & Ackerman (2008). We also show empirically on a large number of datasets with a variety of distance functions that clustering-NSM correlates strongly with ANNS accuracy.

Our second and main contribution is a new clusterability measure that targets clustering-NSM, so that higher clusterability implies a larger clustering-NSM. Our clusterability measure is a statistic summarizing the distribution of point-NSMs, where the point-NSM of a point is the set-NSM of its $r$ nearest neighbors. We prove the positive correlation between point-NSM and clustering-NSM, and show empirically that the relationship holds on a number of datasets with a mix of distance functions.

Our measures inch closer to the ideal we described earlier: A higher point-NSM of a set of points (clusterability) suggests a higher clustering-NSM of a flat clustering of the same points (internal quality measure), which suggests a higher ANNS accuracy (external quality measure). Ours are the first measures to serve as a tool for quantifying the amenability of a set of points to clustering-based ANNS, thereby closing an existing gap in that literature.

## 2 RELATED WORK

Internal clustering quality measures have a long history in the literature. We focus on flat clusterings in our review of the literature because our work is limited to such algorithms in scope. We note, however, that there exist many such measures for hierarchical clustering (Arias-Castro & Coda, 2025), linkage-based clustering (Ben-David & Ackerman, 2008), and others.

The DUNN index (Dunn, 1974) is the ratio of the minimal inter-cluster distance to the maximal intra-cluster distance. Specific instances of this measure are characterized by how they define distances. Typical formulations of intra-cluster distance of a cluster are: the maximum distance between its points, and mean distance between all pairs of points. The inter-cluster distance similarly has different flavors including the distance between centroids, distance between the closest points, and so on.

The measure of Davies & Bouldin (1979) (DB) starts by calculating the following statistic for each cluster. For cluster $i$, it computes the maximal (over all $j \neq i$) ratio between the intra-cluster distances of clusters $i$ and $j$, and the inter-cluster distance between the two clusters. It then reports the mean of these statistics across all clusters. A well-separated cluster has a larger inter-cluster distance to other clusters relative to its intra-cluster distance, so that a lower DB is indicative of better clustering.

There exist other quality measures that are based on similar ideas but are computationally inefficient, such as the widely-used Silhouette coefficient (Rousseeuw, 1987). Relative and Additive Margin (Ben-David & Ackerman, 2008), too, are highly inefficient with a complexity that is exponential in the number of clusters. They are based on the idea that in a better clustering, a point's distance to a representative from its cluster is smaller than its distance to the closest representative from another cluster. The measure reports the minimal mean ratio over all possible representative sets.

All these measures require the distance function to be nonnegative, preventing their application to distances that are functions of inner product. Additionally, they are highly sensitive to the value of the distance function itself, making them susceptible to distortion by the presence of outliers. In contrast, our proposed measure does not consider the magnitude of distances and is not affected by outliers.

Finally, the closest work to ours and a work that our method extends is the notion of $k$-NN consistency developed by Ding & He (2004). As noted earlier, a set is considered $k$-NN consistent if the $k$ nearest neighbors of every point in the set also belong to the set. A clustering measure is then defined as the proportion of $k$-NN consistent clusters. While Ding & He (2004) allow for what they call "fractional" $k$-NN consistency, they do not analyze the method theoretically and on the tasks we evaluate in this work. Additionally, we extend their research to introduce a measure of clusterability.

Clusterability, too, has been extensively studied in the literature. Adolfsson et al. (2019) provide a comprehensive review of the literature and offer an extensive comparative, empirical study of the existing methods on synthetic and real datasets. Ackerman & Ben-David (2009) formalize the problem of clusterability and theoretically analyze existing measures. We refer the interested reader to these two works for a more detailed overview of the literature.

## 3 NEIGHBORHOOD STABILITY MEASURE

We present a detailed description of our proposed methodology. We begin with key concepts and conclude with theoretical results for the proposed clustering quality and clusterability measures.

### 3.1 DEFINITIONS

We write $(\mathcal{X}, \delta)$ to describe a finite set $\mathcal{X} \subset \mathbb{R}^d$ equipped with distance function $\delta : \mathbb{R}^d \times \mathbb{R}^d \to \mathbb{R}$. We denote by $r\text{-NN}_{(\mathcal{X},\delta)}(u)$, $1 \leq r \in \mathbb{Z}$, the $r$ nearest neighbors of the point $u$ in $\mathcal{X}$ with distance function $\delta$: $r\text{-NN}_{(\mathcal{X},\delta)}(u) = \arg\min^{(r)}_{v \in \mathcal{X}, v \neq u} \delta(u, v)$. We drop the subscript and write $r\text{-NN}(\cdot)$ when the set and distance function are clear from the context. We write $\text{NN}(\cdot)$ when $r = 1$.

Writing $|\cdot|$ as the size of a set, we define the following concept:

**Definition 1** ($\alpha$-Stability and Set-Neighborhood Stability Measure)**.** *A set $S$ is $\alpha$-stable, $\alpha \in [0, 1]$, in $(\mathcal{X}, \delta)$ if $S \subseteq \mathcal{X}$ and $|\{u \in S : \text{NN}_{(\mathcal{X},\delta)}(u) \in S\}| = \alpha|S|$. We call $\alpha$ its set-NSM, write set-$\text{NSM}_{(\mathcal{X},\delta)}(S) = \alpha$, and drop the subscript if $(\mathcal{X}, \delta)$ is clear from the context.*

We call $S$ *stable* if it is 1-stable and *unstable* if it is 0-stable. Intuitively, a stable neighborhood forms a tightly-knit cluster that is isolated from the rest of $\mathcal{X}$.

Let us elaborate the connection between $\alpha$-stability and the related notion of $k$-NN consistency (Ding & He, 2004). A set $S$ is said to be $k$-NN consistent if the $k$ nearest neighbors of every point in $S$ also belong to $S$, and inconsistent otherwise. In defining $\alpha$-stability, we relax the definition so that it becomes non-binary, and restrict its scope to $k = 1$. In the terminology of this work then, a $k$-NN consistent set is stable, but a set that is 1-NN-inconsistent can be $\alpha$-stable for some $\alpha \in [0, 1)$.

Let us now extend the notion of set-NSM to a clustering of $\mathcal{X}$.

**Definition 2** (Clustering-Neighborhood Stability Measure). *For a clustering $C = \{C_i\}_{i=1}^L$ of $\mathcal{X}$ such that $\cup_{i=1}^L C_i = \mathcal{X}$ and a set of positive real weights $\omega = \{\omega_i\}_{i=1}^L$, we define the clustering-NSM of $C$ as follows: clustering-$\mathrm{NSM}_\delta(C;\ \omega) = \frac{1}{\sum \omega_i} \sum_{i=1}^L \omega_i$ set-$\mathrm{NSM}(C_i)$. We drop the subscript if $\delta$ is clear from the context.*

In effect, the clustering-NSM of $C$ is the weighted mean of the set-NSM of the clusters that make up $C$. In our experiments to be presented later, we only consider weights that are proportional to the size of each cluster, $\omega_i = |C_i|$, and leave an exploration of other weight schemes to future work.

We claim and empirically show that the NSM of a clustering is an internal clustering quality measure of flat clustering algorithms. But we also present a measure of *clusterability* of a set such that a highly *clusterable* set will have high clustering-NSM. To that end, we define the following:

**Definition 3** (Point-Neighborhood Stability Measure). *A point $u \in \mathcal{X}$ has point-NSM $\alpha$ with radius $r$ if set-$\mathrm{NSM}((r-1)\text{-}\mathrm{NN}(u) \cup \{u\}) = \alpha$. We denote it by point-$\mathrm{NSM}_{(\mathcal{X}, \delta)}(u;\ r)$, but drop the subscript if $(\mathcal{X}, \delta)$ is clear from the context.*

In other words, the point-NSM of $u$ is the set-NSM of the set consisting of $u$ and its $(r-1)$ nearest neighbors in $(\mathcal{X}, \delta)$.

### 3.2 CLUSTERING-NSM AS AN INTERNAL MEASURE OF CLUSTERING QUALITY

Our first result shows that clustering-NSM satisfies the Ben-David & Ackerman (2008) axioms, and as such is a valid internal measure of clustering quality within that framework.

**Theorem 1.** *For a clustering $C$ of $(\mathcal{X}, \delta)$ and a fixed set of weights $\omega$, clustering-$\mathrm{NSM}(C;\ \omega)$ satisfies the four axioms of Ben-David & Ackerman (2008): consistency, richness, scale invariance, and isomorphism invariance. As such, clustering-$\mathrm{NSM}(C; \omega)$ is a measure of clustering quality.*

*Proof.* Let us consider each property separately. In what follows, we write $u \sim_C v$ to indicate that there exists a cluster $C_i$ in the clustering $C = \{C_i\}_{i=1}^L$ such that $u, v \in C_i$. We denote by $u \not\sim_C v$ the alternative case, that $u \in C_i$ and $v \in C_j$ for $i \neq j$.

**Consistency**: Given a clustering $C$ over $(\mathcal{X}, \delta)$, a distance function $\delta'$ is $C$-consistent with $\delta$ if $\delta'(u, v) \leq \delta(u, v)$ for all $u \sim_C v$, and $\delta'(u, v) \geq \delta(u, v)$ for all $u \not\sim_C v$. The *consistency* axiom applied to clustering-NSM requires that, clustering-$\mathrm{NSM}_{\delta'}(C;\ \omega) \geq$ clustering-$\mathrm{NSM}_\delta(C;\ \omega)$ whenever $\delta'$ is $C$-consistent with $\delta$. That holds trivially because for each cluster $C_i \in C$, set-$\mathrm{NSM}_{(\mathcal{X}, \delta')}(C_i) \geq$ set-$\mathrm{NSM}_{(\mathcal{X}, \delta)}(C_i)$.

**Richness**: The *richness* axiom requires that for any non-trivial clustering $C$ of $\mathcal{X}$, there exist a $\delta$ such that $C = \arg\max_{C' \in \mathcal{C}}$ clustering-$\mathrm{NSM}(C';\ \omega)$, where $\mathcal{C}$ is the set of all possible clusterings. This is trivially the case by defining $\delta(u, v) = 1$ for all $u \sim_C v$, and $\delta(u, v) = 10$ for all $u \not\sim_C v$.

**Scale invariance**: This axiom requires that clustering-$\mathrm{NSM}_\delta(C;\ \omega) =$ clustering-$\mathrm{NSM}_{\lambda\delta}(C;\ \omega)$ for all $\lambda > 0$, where $\lambda\delta(u, v) = \lambda \cdot \delta(u, v)$. That holds because $\mathrm{NN}_{(\mathcal{X}, \lambda\delta)}(u) = \mathrm{NN}_{(\mathcal{X}, \delta)}(u)$ for all $u \in \mathcal{X}$, so that set-$\mathrm{NSM}_{(\mathcal{X}, \lambda\delta)}(C_i) =$ set-$\mathrm{NSM}_{(\mathcal{X}, \delta)}(C_i)$ for all $C_i \in C$.

**Isomorphism invariance**: Two clusterings $C$ and $C'$ over $(\mathcal{X}, \delta)$ are said to be isomorphic, denoted $C \approx_\delta C'$, if there exists a distance-preserving isomorphism $\phi$ such that for all $u, v \in \mathcal{X}$, $u \sim_C y$ iff $\phi(u) \sim_{C'} \phi(v)$. The *isomorphism invariance* property requires that clustering-$\mathrm{NSM}(C;\ \omega) =$ clustering-$\mathrm{NSM}(C';\ \omega)$ for all $C \approx_\delta C'$. But that holds because $\phi$ is distance-preserving, implying that $\mathrm{NN}_{(\mathcal{X}, \delta)}(u) = \mathrm{NN}_{(\phi(\mathcal{X}), \delta)}(\phi(u))$ for all $u \in \mathcal{X}$, where we used the short-hand $\phi(\mathcal{X}) = \{\phi(u);\ u \in \mathcal{X}\}$. □

We close this section by a note on the efficiency of clustering-NSM. Given the nearest neighbors of all points in $\mathcal{X}$, clustering-NSM can be computed in time linear in the size of $\mathcal{X}$. Importantly, $\text{NN}(u)$ for any point $u \in \mathcal{X}$ is independent of the clustering $C$ and can be computed upfront and reused to evaluate any clustering of the set.

For large sets, the complexity of finding the nearest neighbor of each point in a set of $n$ points, which is $\mathcal{O}(n^2)$, can be improved to $\mathcal{O}(n^{3/2})$ or $\mathcal{O}(n \log n)$ if one replaces the exact nearest neighbor subroutine with its approximate variant (Bruch, 2024) in Definition 1 to compute stability measures. We will later show that, in practice, such approximation makes no perceptible difference in the clustering-NSM.

### 3.3 POINT-NSM AS A MEASURE OF CLUSTERABILITY

We now claim that point-NSM is a measure of clusterability, so that a set whose clusterability is higher according to point-NSM yields a clustering with a larger clustering-NSM. Intuitively, we argue that, if the distribution of point-NSM is concentrated around a large value, then its clustering-NSM is also large. We prove that claim for *spherical* flat clusterings in the next result, and extend the result to the more general case in Appendix A.

**Theorem 2.** *Consider $(\mathcal{X}, \delta)$ in $\mathbb{R}^d$ where $|\mathcal{X}| = L \cdot r$ for integers $L, r > 1$. Suppose that $\mathcal{X}$ can be clustered into $L$ $\delta$-balls each centered at a point $u \in \mathcal{X}$ and consisting of $r$ points. Suppose that each point $u$ is equally likely to be at the center of a ball. Denote the set of all such clusterings by $\mathcal{C}$. If $\omega_i = r$, then $\mathbb{E}_{C \sim \mathcal{C}}[\text{clustering-NSM}(C; \omega)] = \mathbb{E}_{u \sim \mathcal{X}}[\text{point-NSM}(u; r)]$. Furthermore, clustering-NSM$(C; \omega) \leq \mathbb{E}[\text{point-NSM}(u; r)] - \sqrt{\frac{\log(1/\epsilon)}{2L}}$ with probability at most $\epsilon$.*

*Proof.* For each $u \in \mathcal{X}$, define $B_r(u)$ as the ball centered at $u$ with radius $\max_{v \in (r-1)\text{-NN}(u)} \delta(u, v)$. From all $L \cdot r$ such balls, find all non-overlapping subsets that cover $\mathcal{X}$. The set of covers make up $\mathcal{C}$. Each ball in $C \sim \mathcal{C}$ has set-NSM equal to the point-NSM of the point at its center: set-NSM$(B_r(u)) = $ point-NSM$(u; r)$ for all $u$ such that $B_r(u) \in C$. Now, consider:

$$\mathbb{E}_{C \sim \mathcal{C}}[\text{clustering-NSM}(C; \omega)] = \mathbb{E}[\frac{1}{L} \sum_{B_r(u) \in C} \text{set-NSM}(B_r(u))] = \frac{1}{L} \mathbb{E}[\sum_{u: B_r(u) \in C} \text{point-NSM}(u; r)]$$

$$= \frac{1}{L} \mathbb{E}[\sum_{u \in \mathcal{X}} \mathbb{1}_{B_r(u) \in C} \text{point-NSM}(u; r)] = \frac{1}{L} \sum_{u \in \mathcal{X}} \mathbb{E}[\text{point-NSM}(u; r) \mathbb{1}_{B_r(u) \in C}]$$

$$= \frac{1}{L \cdot r} \sum_{u \in \mathcal{X}} \text{point-NSM}(u; r) = \mathbb{E}_{u \sim \mathcal{X}}[\text{point-NSM}(u; r)].$$

Notice that, while the indicator random variables $\mathbb{1}_{B_r(u) \in C}$ are dependent on each other (because choosing one point changes the probability of others being selected), they are exchangeable. That is because, as long as $L$ balls are selected, it does not matter which individual balls those are. As such, Hoeffding's bounds for iid variables hold (Barber, 2025), giving the required bound. $\square$

We note that, while the conditions of this result are rather strong, as we demonstrate later, the result holds even when these conditions may not be fully satisfied. We also highlight that $\delta$ can be any distance function for which a clustering of the data is well-defined. Notably, this includes Euclidean and angular distances, but not inner product. As we see later, however, the result holds empirically even for inner product.

## 4 EMPIRICAL EVALUATION OF CLUSTERING-NSM

We now put Theorem 1 to the test by applying it to clustering-based ANNS. To show the generalizability of our measures, we also apply them to the task of image clustering. For each task, we describe the datasets and the task, and discuss the evaluation protocol, our hypothesis, and empirical findings. All our code and data required to reproduce our experiments can be found in a double-blind review-compliant Git repository.[2]

---

[2] https://anonymous.4open.science/r/nsm-5DEC/README.md

In addition to clustering-NSM, we evaluate the DUNN index (Dunn, 1974) as well as the Davies-Bouldin (DB) index (Davies & Bouldin, 1979) as popular representatives of internal measures of clustering quality. However, to make the DB index more directly comparable with clustering-NSM, we introduce a weighted variant as follows:

$$\mathrm{DB}_\delta(C; \omega) = \frac{1}{\sum \omega_i} \sum_{i=1}^{L} \omega_i \max_{j \neq i} \big( \frac{\sigma_i + \sigma_j}{\delta(\mu_i, \mu_j)} \big), \tag{1}$$

where $\delta$ is a *nonnegative* distance function (typically, Euclidean), and $\sigma_i$ the average distance of points in $C_i$ to their mean $\mu_i$. When $\omega_i = 1$ for all $i$'s, the weighted DB coincides with its original definition, but we also let $\omega_i = |C_i|$ to ensure that the weighting of clustering-NSM is not the sole factor responsible for its performance compared with vanilla DB.

Finally, we leave out other internal measures of clustering quality (e.g., the Silhouette coefficient and Relative Margin) as they often do not easily scale to the dataset sizes we consider in our experiments.

## 4.1 CLUSTERING-BASED ANNS

**Task**: As noted earlier, nearest neighbor (NN) search solves the following problem for an arbitrary query vector, $q \in \mathbb{R}^d$:

$$\underset{u \in \mathcal{X}}{\arg\min}^{(k)} \quad \delta(q, u), \tag{2}$$

where the superscript $(k)$ indicates that $\arg\min$ returns the $k$ minimizers of its argument. Different instances of this problem are characterized by the distance (or similarity) function. Solving the exact NN search problem can be computationally expensive, however. As such, one often resorts to an approximate variant of the problem known as ANNS (Bruch, 2024), whose accuracy is quantified as follows: if $S'$ is the set of $k$ points returned by an ANNS algorithm and $S$ is the exact set of $k$ points from an NN search algorithm, accuracy is simply $|S \cap S'|/k$.

As we noted before, in clustering-based ANNS—also known as Inverted File (IVF) (Jégou et al., 2011)—a dataset is first partitioned into $L$ shards using a clustering algorithm such as KMEANS. Search involves two stages. First, the algorithm *routes* the query $q$ to $\ell \ll L$ shards, where $\ell$ is a hyperparameter. Typically, these are the $\ell$ shards whose centroids minimize $\delta(q, \cdot)$, though other more complex routing subroutines exist (Guo et al., 2020; Bruch et al., 2024a; Vecchiato et al., 2024; Gottesbüren et al., 2024). In a subsequent stage, NN search (or ANNS) is performed over just the $\ell$ shards. In this formulation, $\ell$ (also called `nprobe`) trades off accuracy for speed.

**Datasets**: We use a suite of benchmark datasets from the ANNS literature and challenges (Simhadri et al., 2022; 2024). The datasets cover all three common distance or similarity functions (Euclidean, cosine similarity, and inner product), as well as a range of collection sizes and dimensionalities.

We give a complete description of the datasets in Appendix B. What is most relevant is the distance function, and size and dimensionality of the datasets which we note briefly here. For Euclidean search: MNIST and FASHION-MNIST (60,000 points, $d = 784$); GIST (100,000, $d = 960$); and, SIFT (1m, $d = 128$). For Cosine Similarity search: DEEPIMAGE (10m, $d = 96$); GLOVE (1.2m, $d = 200$); LASTFM (300,000, $d = 65$); MSMARCO (8.8m, $d = 384$); and, NQ (2.7m, $d = 1{,}536$). For Maximum Inner Product Search: TEXT2IMAGE (10m, $d = 200$); and, MUSIC (1m, $d = 100$).

**Evaluation protocol**: We cluster a given dataset $\mathcal{X}$ into $L = \sqrt{|\mathcal{X}|}$ clusters to prepare for clustering-based ANNS using a number of clustering algorithms to be described shortly. In Appendix D we also experiment with $L = t\sqrt{|\mathcal{X}|}$ for $t \in \{1/4, 1/2\}$ for completeness. We fix $\ell = 1$ and report results for $k \in \{5, 10\}$—we study the effect of $\ell$ in Appendix E. For each $k$ and a clustering $C$, we measure ANNS accuracy, and report the Spearman's rank correlation coefficient between accuracy and the clustering quality measures.

As for clustering, we use standard and spherical variants of KMEANS. We run each for $\{5, 10, 20, 40\}$ iterations, resulting in a total of 8 clusterings with more iterations leading to better quality. We note that, due to scalability issues, we are unable to conduct experiments with other flat clustering algorithms such as DBScan (Ester et al., 1996) or spectral clustering (Ng et al., 2001).

**Hypothesis**: For a fixed algorithm, a larger number of iterations leads to better clustering, which in turn improves ANNS accuracy. Furthermore, as evidenced by past research (Bruch et al., 2024a),

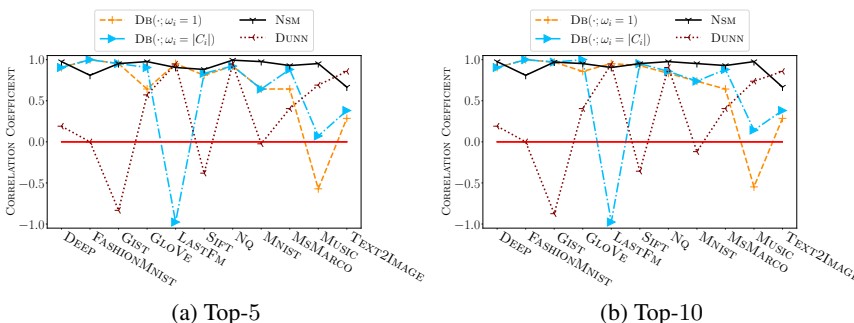

(a) Top-5           (b) Top-10

Figure 1: Spearman's correlation coefficient between clustering quality and top-$k$ ANNS accuracy. For DB, we present the negated coefficient because smaller values indicate better clustering quality.

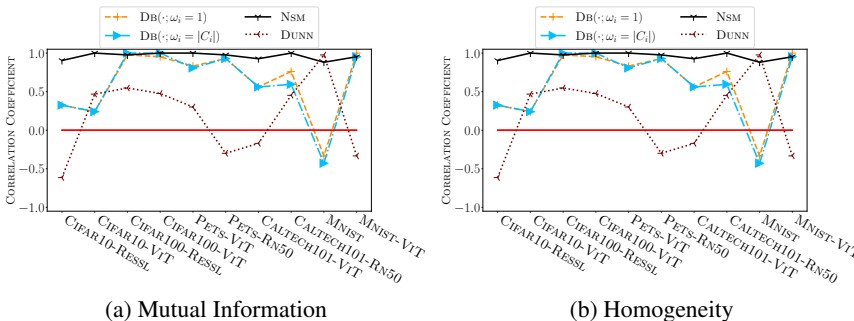

(a) Mutual Information           (b) Homogeneity

Figure 2: Spearman's correlation coefficient between clustering quality and external evaluation metrics. For DB, the coefficient is negated because a smaller index indicates better clustering quality.

spherical KMEANS outperforms standard KMEANS for certain distance functions. Our expectation is that internal quality measures should strongly correlate with ANNS accuracy, given the 8 accuracy and quality measurement pairs, with the null hypothesis that no strong correlation can be observed.

**Results**: Figure 1 renders the results of our experiments. The plots illustrate Spearman's correlation coefficient between ANNS accuracy for top-$k$ search ($k \in \{5, 10\}$) and clustering quality measures.

Observe that the DUNN index is not predictive of ANNS accuracy, and swings wildly from one dataset to another. The two variants of the DB index show relatively stronger correlation, but weaken for several datasets. That is particularly the case when ANNS is by inner product; in fact, on the MUSIC dataset, there is (weak) anti-correlation between DB and accuracy.

Clustering-NSM (denoted NSM in the figure for brevity) shows a stronger correlation, even on inner product tasks. We note that, in Figure 1, we use exact nearest neighbors in Definition 1 to calculate clustering-NSM. In Figure 8 in Appendix F, we present results for a configuration where we use approximate nearest neighbors instead.

The empirical observations provide sufficient evidence that clustering-NSM is a strong predictor of ANNS accuracy. Given the statistically significant differences at $k = 10$ ($p$-value being smaller than 0.001), we can safely reject the null hypothesis. The same cannot be concluded for other measures.

## 4.2 IMAGE CLUSTERING

**Task**: In this task, we wish to cluster a set of images into categories in an unsupervised manner. One approach to this task is what is known as "deep clustering," (Zhou & Zhang, 2022) consisting of representation learning using a deep network, followed by the application of a learned clustering method. In our experiments, we apply unsupervised clustering to learned representations of data.

**Datasets**: We use a suite of benchmark datasets, encoded using state-of-the-art vision models. As before, in choosing these datasets, we intend to cover a range of dataset sizes and dimensionality.

We give a full description of the datasets in Appendix C, but include a brief summary here: CIFAR10 and CIFAR100 (10 and 100 classes) embedded with CLIP-VIT-L14 (Radford et al., 2021) and RESSL (He et al., 2016), denoted by "VIT" and "RESSL" suffixes; MNIST-VIT and MNIST-RESSL (10 classes); PETS-VIT (37 classes) embedded with CLIP-VIT-L14 and PETS-RN50 embedded with CLIP-RESNET50 (He et al., 2016); and, CALTECH101-VIT and CALTECH101-RN50 (101 classes).

**Evaluation protocol**: As in the ANNS experiments, we cluster a dataset into the target number of classes associated with the dataset (e.g., 10 for CIFAR10 and MNIST). Once clustered, we measure *external evaluation* metrics including Mutual Information and Homogeneity, and report the Spearman's correlation coefficient between each metric and clustering quality. As before, we use standard and spherical KMEANS, with 5, 10, 20, and 40 iterations, resulting in 8 different clusterings.

**Hypothesis**: A larger number of iterations of a fixed algorithm leads to better clustering, which in turn improves Mutual Information and Homogeneity. Our expectation is that quality measures should strongly correlate with the metrics, given the 8 external metric and quality measurement pairs. The null hypothesis once again states that no strong correlation can be observed.

**Results**: We present the results of our experiments in Figure 2. Each figure plots the Spearman's correlation coefficient between one of the external evaluation metrics and clustering quality measures.

Neither the DUNN nor the DB index is consistently predictive of external evaluation metrics. Clustering NSM, however, shows a strong correlation with both Mutual Information and Homogeneity. All of clustering-NSM's measured correlation coefficients are statistically significant (with $p$-values less than 0.001), rejecting the null hypothesis. As with the ANNS task in Section 4.1, we present in Appendix F results where we use approximate nearest neighbors to compute stability measures.

## 5 EMPIRICAL EVALUATION OF POINT-NSM FOR CLUSTERABILITY

This section tests Theorem 2 on datasets from Sections 4. We fix $r$ from the set $\{2^{10}, 2^{11}, 2^{12}, 2^{13}\}$. We then extract the distribution of point-NSMs with radius $r$ for each dataset $(\mathcal{X}_i, \delta_i)$ and choose its mean or an $\alpha$-quantile (e.g., $\alpha = 0.1$) as a summary statistic of the distribution, denoted $s_i$. For scalability, we only consider 5% of points from each dataset uniformly at random to compute the point-NSM distribution—Appendix G presents results without sub-sampling. Next, we cluster (all of) $\mathcal{X}_i$ into $L_i = |\mathcal{X}_i|/r$ clusters using standard or spherical KMEANS and compute their clustering-NSM.

Given all pairs of $(s_i, \text{clustering-NSM}_{\delta_i}(C^{(i)}; \omega^{(i)}))$, where $\omega_j^{(i)} = |C_j^{(i)}|$ for the $j$-th cluster of the $i$-th dataset, we report the Spearman's correlation coefficient. Theorem 2 suggests that this correlation should be strong, but we note that the requirement of the theorem is not guaranteed to hold.

Figure 3 shows the distribution of point-NSMs for all datasets and $r = 1024$ and $r = 8192$. Unsurprisingly, as $r$ increases, the point-NSM distribution shifts to the right—that trend holds across all four values of $r$. However, it is clear that different datasets have different point-NSM distributions.

The differences between these distributions should, per Theorem 2, be reflected in clustering-NSMs. We observe that to be true: Table 1 shows that point-NSM statistics (mean and 0.1-quantile) correlate strongly with clustering-NSMs. Unsurprisingly, as the quantile approaches 1, correlation weakens (excluded from Table 1). Quantiles smaller than 0.5 perform similarly to those reported in the table.

## 6 DISCUSSION

We learned from Section 4 that clustering-NSM is an appropriate quality measure for tasks that involve the application of flat clustering. Examples include clustering-based ANNS—which organizes points into clusters for faster albeit approximate search—and classification via clustering—which requires extraction of geometric structures from a set of points. Importantly, clustering-NSM performs more strongly than other measures examined in this work, determined by the correlation between the internal quality measures and the task's external measures of quality.

That result is not surprising in and of itself. What is perhaps more interesting—and one of the novel contributions of this work—is the notion of point-NSM as a measure of *clusterability*. In particular, the distribution of point-NSMs correlates with the clustering-NSM of a flat clustering. Putting together

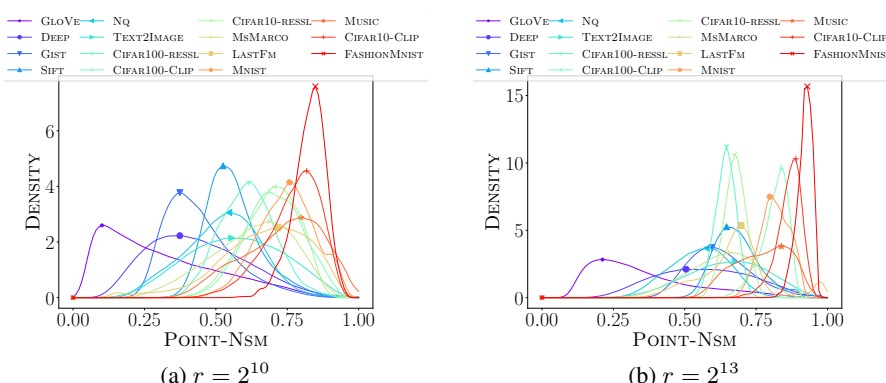

(a) $r = 2^{10}$ (b) $r = 2^{13}$

Figure 3: Point-Nsm distributions for various datasets.

Table 1: Spearman's correlation coefficient between statistics from point-Nsm distributions and the clustering-Nsm for clusterings obtained from standard and spherical KMeans. All correlations are significant ($p$-value $< 0.001$). Last row aggregates statistics and clustering-Nsm for all values of $r$.

| POINT-NSM RADIUS | STANDARD KMEANS | | SPHERICAL KMEANS | |
|---|---|---|---|---|
| | MEAN | 0.1-QUANTILE | MEAN | 0.1-QUANTILE |
| $r = 1024$ | 0.81 | 0.86 | 0.79 | 0.85 |
| $r = 2048$ | 0.82 | 0.85 | 0.82 | 0.85 |
| $r = 4096$ | 0.74 | 0.81 | 0.74 | 0.81 |
| $r = 8192$ | 0.76 | 0.78 | 0.76 | 0.78 |
| COMBINED | 0.79 | 0.83 | 0.82 | 0.85 |

the two pieces, we have presented a measure of clusterability that correlates with clustering-Nsm, which in turn correlates with task performance metrics.

This reliable way of assessing if a dataset is clusterable bridges a gap that has existed in the Anns literature: Determining whether clustering-based Anns is suitable for a dataset can now be done by measuring its point-Nsm distribution. This is of particular interest because point-Nsm is *independent* of the query distribution. Contrast that with existing evaluation protocols that simply use a test query distribution to determine the efficacy of search on a dataset (Vecchiato et al., 2024; Bruch et al., 2024b; Douze et al., 2024).

## 7 CONCLUDING REMARKS

In this work, we presented an internal measure of clustering quality, called clustering-Nsm, that is a function of the nearest neighbor stability of each cluster. We further showed that the same concept can be used as a measure of clusterability of a collection of points, which we referred to as point-Nsm. Empirically, a dataset with a distribution of point-Nsms that is concentrated closer to 1 has a higher clustering-Nsm for flat clustering algorithms. A higher clustering-Nsm, in turn, implies a higher performance metric for an end-task such as Anns and classification.

Our measures are agnostic to the distance function, and extend to nonnegative functions such as inner product. However, they target tasks that depend on the geometric structure of the data. That family includes all flavors of Anns, as well as classification tasks where the geometry of the point collection carries information about ground-truth labels.

We leave a few questions as future work. We wish to expand our understanding of the role of weights in Definition 2, which also affects Theorem 2. We also plan to investigate if point-Nsm can provide a theoretical tool for explaining the efficacy of widely-used graph-based Anns algorithms.

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

## A  EXTENSION OF THEOREM 2

In Section 3.3, we showed that the average point-NSM of a dataset coincides with the average clustering-NSM over spherical flat clusterings of the same data. In this section, we consider general flat clusterings (i.e., spherical or non-spherical) and show that the result holds with a slight modification.

**Theorem 3.** *Consider $(\mathcal{X}, \delta)$ in $\mathbb{R}^d$ where $|\mathcal{X}| = L\ell r$ for integers $L, \ell, r > 1$. Suppose that $\mathcal{X}$ can be clustered into $L$ clusters of size $\ell r$ each, where a cluster can be decomposed into a set of $\ell$ $\delta$-balls each consisting of $r$ points. Denote the set of all such clusterings by $\mathcal{C}$. If $\omega_i = \ell r$, then $\mathbb{E}_{C \sim \mathcal{C}}[clustering\text{-}\mathrm{NSM}(C; \omega)] \geq \mathbb{E}_{u \sim \mathcal{X}}[point\text{-}\mathrm{NSM}(u; r)]$. Furthermore, $clustering\text{-}\mathrm{NSM}(C; \omega) \leq \mathbb{E}[point\text{-}\mathrm{NSM}(u; r)] - \sqrt{\frac{\log(1/\epsilon)}{2L}}$ with probability at most $\epsilon$.*

*Proof.* Our argument closely follows the proof of Theorem 2. Define as before $B_r(u)$ to be a ball centered at $u$ with radius $\max_{v \in (r-1)\text{-}\mathrm{NN}(u)} \delta(u, v)$.

First, observe that for a single cluster $C_i$, there is by assumption a set of points $\hat{C}_i = \{u_j\}_{j=1}^{\ell} \subset C_i$ such that $C_i = \cup_{u \in \hat{C}_i} B_r(u)$. Clearly:

$$\text{set-}\mathrm{NSM}(C_i) \geq \frac{1}{\ell} \sum_{u \in \hat{C}_i} \text{set-}\mathrm{NSM}(B_r(u)).$$

Let us now construct $\mathcal{C}$. Form $B_r(u)$ for every $u \in \mathcal{X}$. From all $L \cdot \ell \cdot r$ such balls, find $L$ non-overlapping subsets that cover $\mathcal{X}$. Each $L$ non-overlapping subset is a valid clustering, so that the set of subsets make up $\mathcal{C}$. We can now state the following:

$$\mathbb{E}_{C \sim \mathcal{C}}[\text{clustering-}\mathrm{NSM}(C; \omega)] = \mathbb{E}[\frac{1}{L} \sum_{C_i \in C} \text{set-}\mathrm{NSM}(C_i)] \geq \mathbb{E}[\frac{1}{L} \sum_{C_i \in C} \frac{1}{\ell} \sum_{u \in \hat{C}_i} \text{set-}\mathrm{NSM}(B_r(u))].$$

Notice that $\mathcal{X} = \cup_{i=1}^{L} C_i = \cup_{i=1}^{L} \cup_{u \in \hat{C}_i} B_r(u)$. Denoting by $\hat{C} = \cup_{i=1}^{L} \hat{C}_i$, we can write $\mathcal{X} = \cup_{u \in \hat{C}} B_r(u)$. We have effectively reduced the problem to the setup of Theorem 2, where a clustering is assumed to be made up of $\delta$-balls. Therefore, the right-hand-side is equal to the average point-NSM, giving:

$$\mathbb{E}_{C \sim \mathcal{C}}[\text{clustering-}\mathrm{NSM}(C; \omega)] \geq \mathbb{E}_{u \sim \mathcal{X}}[\text{point-}\mathrm{NSM}(u; r)].$$

We have thus bounded the expected value of clustering-NSM from below, which enables us to apply the one-sided Hoeffding's inequality to obtain a conservative bound on its value:

$$\begin{aligned}
\exp\left(-2Lt^2\right) &\geq \mathbb{P}\left[\text{clustering-}\mathrm{NSM}(C; \omega) - \mathbb{E}[\text{clustering-}\mathrm{NSM}(C; \omega)] \leq -t\right] \\
&= \mathbb{P}\left[\text{clustering-}\mathrm{NSM}(C; \omega) - (\mathbb{E}[\text{point-}\mathrm{NSM}(u; r)] + \theta) \leq -t\right] \quad (\theta \geq 0) \\
&= \mathbb{P}\left[\text{clustering-}\mathrm{NSM}(C; \omega) - \mathbb{E}[\text{point-}\mathrm{NSM}(u; r)] \leq -t + \theta\right] \\
&\geq \mathbb{P}\left[\text{clustering-}\mathrm{NSM}(C; \omega) - \mathbb{E}[\text{point-}\mathrm{NSM}(u; r)] \leq -t\right] \\
&= \mathbb{P}\left[\text{clustering-}\mathrm{NSM}(C; \omega) \leq \mathbb{E}[\text{point-}\mathrm{NSM}(u; r)] - t\right].
\end{aligned}$$

Let the probability upper bound be denoted by $\epsilon$:

$$\epsilon = \exp(-2Lt^2) \implies \log(\epsilon) = -2Lt^2 \implies -\log(\epsilon) = 2Lt^2 \implies \log(1/\epsilon) = 2Lt^2$$

$$\implies t^2 = \frac{\log(1/\epsilon)}{2L} \implies t = \sqrt{\frac{\log(1/\epsilon)}{2L}}.$$

Thus, we obtain the following bound on clustering-NSM, which holds with probability at most $\epsilon$:

$$\text{clustering-}\mathrm{NSM}(C; \omega) \leq \mathbb{E}[\text{point-}\mathrm{NSM}(u; r)] - \sqrt{\frac{\log(1/\epsilon)}{2L}}.$$

$\square$

## B FULL DESCRIPTION OF ANN DATASETS

We describe the datasets used in Section 4.1 in more detail here:

- Euclidean Search:

  - MNIST: A collection of 60,000, 28×28 pixel grayscale images of handwritten digits (0 through 9) flattened as 784-dimensional vectors (Deng, 2012). The dataset has 10,000 additional points used as queries. The dataset is made available under the terms of the Creative Commons Attribution-Share Alike 3.0 license.
  - FASHION-MNIST: A dataset (Xiao et al., 2017) similar in size and dimensionality to MNIST, but representing images of 10 fashion categories. The dataset is released under the The MIT License (MIT) Copyright ©2017 Zalando SE, https://tech.zalando.com.
  - GIST: A collection of 100,000, 960-dimensional image descriptors (Oliva & Torralba, 2001) with 1,000 query points. The dataset is available under the terms of the Creative Commons CC0 Public Domain Dedication license.
  - SIFT: A subset of 1 million, 128-dimensional data points along with 10,000 queries (Jégou et al., 2011). The dataset is available under the terms of the Creative Commons CC0 Public Domain Dedication license.

- Cosine Similarity Search:

  - DEEPIMAGE: Subset of 10 million 96-dimensional points from the billion deep image features dataset (Yandex & Lempitsky, 2016) with 10,000 queries. The dataset is available under the terms of Apache license 2.0.
  - GLOVE: 1.2 million, 200-dimensional word embeddings trained on tweets (Pennington et al., 2014) with 10,000 test queries. The dataset is available under the terms of the Public Domain Dedication and license v1.0.
  - LASTFM: A dataset of about 300,000 vectors representing 65 audio features per contemporary song, along with 50,000 query points (Bertin-Mahieux et al., 2018). The dataset is available for research only, strictly non-commercial use as per http://millionsongdataset.com/lastfm/.
  - MSMARCO: MS MARCO Passage Retrieval v1 (Nguyen et al., 2016) is a question-answering text dataset consisting of 8.8 million short passages in English. We use the "dev" set of queries for retrieval, made up of 6,980 questions. We embed individual passages and queries using the ALL-MINILM-L6-V2 model[3] to form a 384-dimensional vector collection. The dataset is available under the terms of the Creative Commons Attribution 4.0 International license. The model used to embed the dataset is available under the terms of Apache license 2.0.
  - NQ: 2.7 million, 1,536-dimensional embeddings of the Natural Questions dataset (Kwiatkowski et al., 2019) with the ADA-002 model.[4] The dataset is available under the terms of Apache license 2.0.

- Maximum Inner Product Search:

  - TEXT2IMAGE: A cross-modal dataset, where data and query points may have different distributions in a shared space (Simhadri et al., 2022). We use a subset consisting of 10 million 200-dimensional data points along with a subset of 10,000 test queries. The dataset is available under the terms of the Creative Commons Attribution 4.0 International license.
  - MUSIC: 1 million 100-dimensional points (Morozov & Babenko, 2018) with 1,000 queries. To the best of our knowledge, the dataset comes with no information on the license under which it is made available (per https://github.com/stanis-morozov/ip-nsw?tab=readme-ov-file).

---

[3]Checkpoint at https://huggingface.co/sentence-transformers/all-MiniLM-L6-v2.

[4]https://openai.com/index/new-and-improved-embedding-model/

## C  FULL DESCRIPTION OF IMAGE DATASETS

The following is a complete description of the datasets used in the experiments Section 2:

- CIFAR10 and CIFAR100: A collection of 60,000, $32 \times 32$ color images in, respectively, 10 and 100 non-overlapping categories (Krizhevsky & Hinton, 2009), available under the terms of The MIT License (MIT) Copyright ©2021 Eka A. Kurniawan. The collections are labeled subsets of the 80 million tiny images dataset (Torralba et al., 2008). We transform these images using the following vision models:
    - CLIP-VIT-L14: A CLIP (Radford et al., 2021) model that uses the Vision Transformer (VIT) (Dosovitskiy et al., 2021) architecture to encode images, available under the terms of MIT License Copyright ©2021 OpenAI. In particular, we use the VIT-L14 model that is trained on 224-by-224 pixel images. We denote this representation of a dataset by appending VIT to the name of the dataset (e.g., CIFAR10-VIT). The vectors produced by this model are in a 768-dimensional space governed by cosine similarity.
    - RESSL: A vision model that is trained using self-supervised learning and encodes images into a 512-dimensional space, where distance is based on cosine similarity (Zheng et al., 2021).
- MNIST: A collection of 70,000, $28 \times 28$ pixel grayscale images of handwritten digits (0 through 9) (Deng, 2012). We represent this dataset by a) flattening the images as 784-dimensional vectors where distance is based on the Euclidean norm (denoted simply by MNIST); b) the CLIP-VIT-L14 and RESSL models as described earlier, denoted by MNIST-VIT and MNIST-RESSL, respectively. The dataset is made available under the terms of the Creative Commons Attribution-Share Alike 3.0 license.
- PETS: An annotated dataset of pets covering 37 different breeds of cats and dogs (Parkhi et al., 2012). We encode the images using CLIP-VIT-L14 and CLIP-RESNET50 (He et al., 2016) (denoted RN50), producing 1024-dimensional vectors. The dataset is available under the terms of the Creative Commons Attribution-ShareAlike 4.0 International license.
- CALTECH101: Collection of pictures of objects from 101 categories (Fei-Fei et al., 2004). We encode the images using CLIP-VIT-L14 and CLIP-RESNET50. The dataset is available under the terms of the Creative Commons CC0 Public Domain Dedication license.

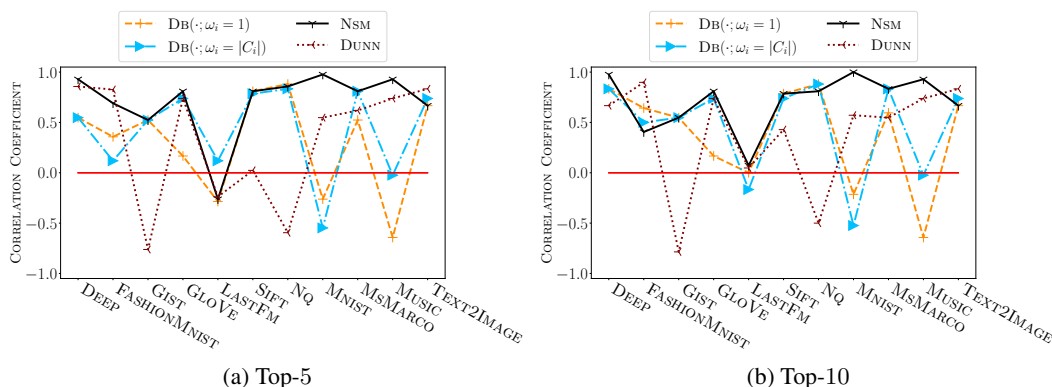

(a) Top-5          (b) Top-10

Figure 4: Spearman's correlation coefficient between clustering quality and top-$k$ ANN accuracy where the number of clusters is $\frac{1}{4}\sqrt{|\mathcal{X}|}$. For DB, we present the negated coefficient because smaller values indicate better clustering quality.

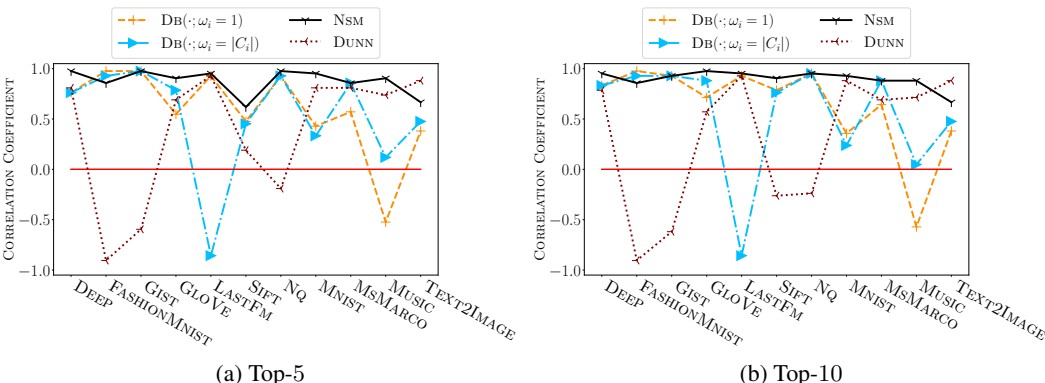

(a) Top-5          (b) Top-10

Figure 5: Spearman's correlation coefficient between clustering quality and top-$k$ ANN accuracy where the number of clusters is $\frac{1}{2}\sqrt{|\mathcal{X}|}$. For DB, we present the negated coefficient because smaller values indicate better clustering quality.

## D   EFFECT OF NUMBER OF CLUSTERS ON THE ANN TASK

In Section 4.1, we set the number of clusters in the clustering-based ANN task to $L = \sqrt{|\mathcal{X}|}$ for dataset $\mathcal{X}$ to form the index. To study the effect of $L$ on the relative performance of different clustering quality measures, we present results for $L = \frac{1}{4}\sqrt{|\mathcal{X}|}$ and $L = \frac{1}{2}\sqrt{|\mathcal{X}|}$ in Figures 4 and 5, respectively. The trends observed in Section 4.1 still hold. However, it should be noted that, when the number of clusters is too small and consequently clusters are much larger, all quality measures' correlation with ANN accuracy degrades significantly. This, however, is not a surprising result.

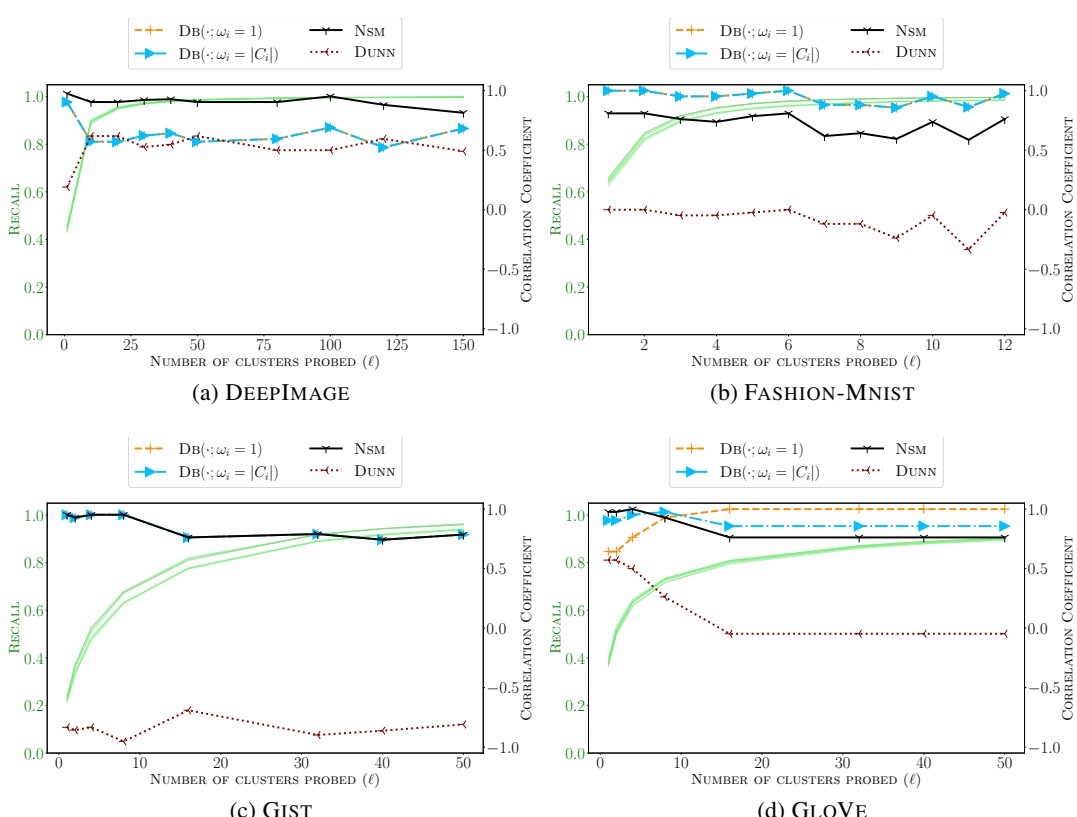

Figure 6: Spearman's correlation coefficient between clustering quality and top-5 ANNS accuracy, as a function of $\ell$ with values between $1$ and $5\%$ of the total number of clusters. The figures also include, for each of the eight clustering algorithms, recall (green curves) versus $\ell$ for reference. For DB, we present the negated coefficient because smaller values indicate better clustering quality.

## E    CLUSTERING QUALITY MEASURES AND NUMBER OF PROBED CLUSTERS

In the experiments presented in Section 4.1, we configured the clustering-based ANNS algorithm to route queries to a single cluster by setting $\ell = 1$. We then computed the correlation between ANNS accuracy and each clustering quality measure, given a set of ANNS indices.

In this appendix, we examine the effect of $\ell$ on correlation coefficients. As $\ell$ becomes larger, ANNS accuracy increases, with accuracy saturating when $\ell$ is large enough relative to the number of clusters $L$. The question we wish to shed light on is whether measures of clustering quality are predictive of the relative performance of ANNS as $\ell$ becomes larger.

We do so by repeating the experiments of Section 4.1 reported in Figure 1, but by setting $\ell$ to a sequence of values—up to roughly $5\%$ of the number of clusters. Figures 6 and 7 plots the correlation coefficients at each $\ell$ for all measures and every dataset. We have also included a plot of ANNS accuracy as a function of $\ell$ for completeness.

We observe that on most datasets correlation coefficients suffer some degradation as $\ell$ grows. However, the relative performance of clustering quality measures remains stable. That is, where clustering-NSM is more correlated with ANNS accuracy than DB at $\ell = 1$, it remains so as $\ell$ grows. The exception to that is GLOVE where clustering-NSM's correlation becomes smaller than DB's when $\ell$ is roughly $1\%$ of the total number of clusters, $L$.

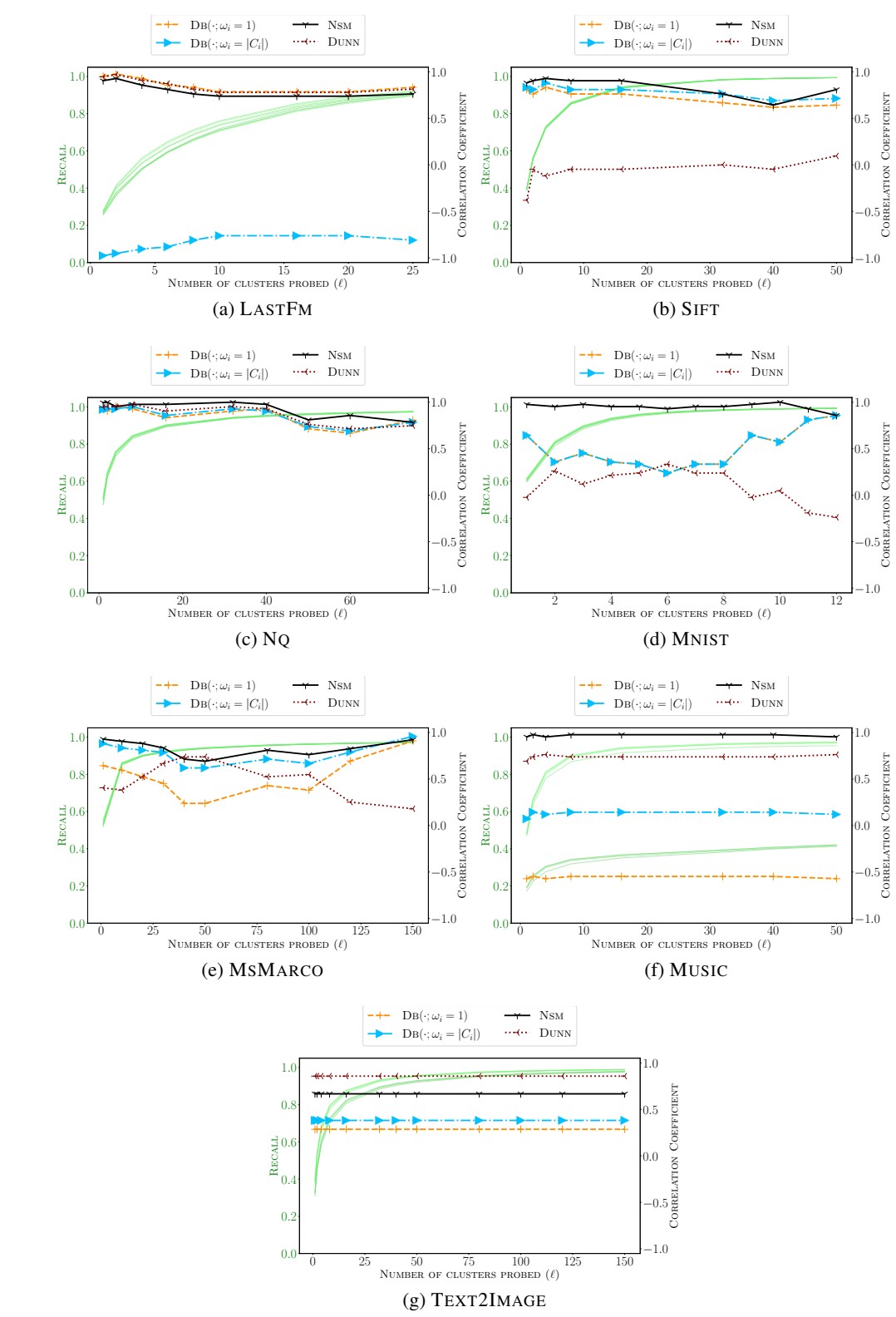

Figure 7: Figure 6 continued.

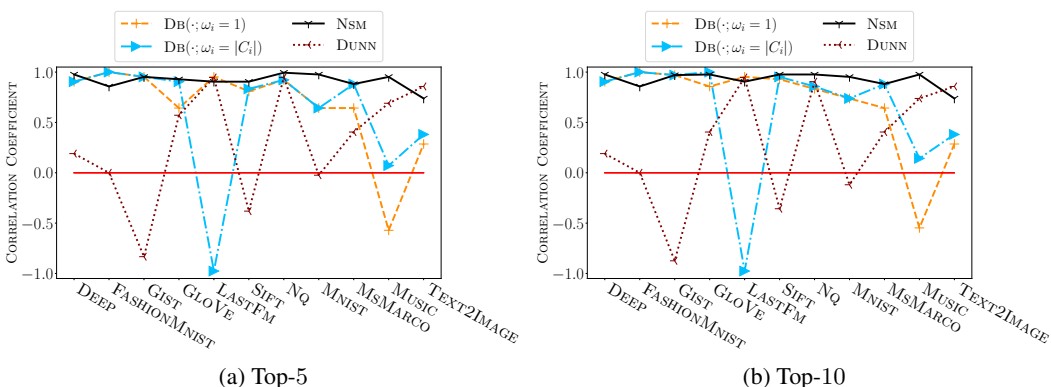

(a) Top-5                (b) Top-10

Figure 8: Spearman's correlation coefficient between clustering quality and top-$k$ ANN accuracy. For DB, we present the negated coefficient because smaller values indicate better clustering quality. For clustering-NSM, we use approximate nearest neighbors to compute each set's stability.

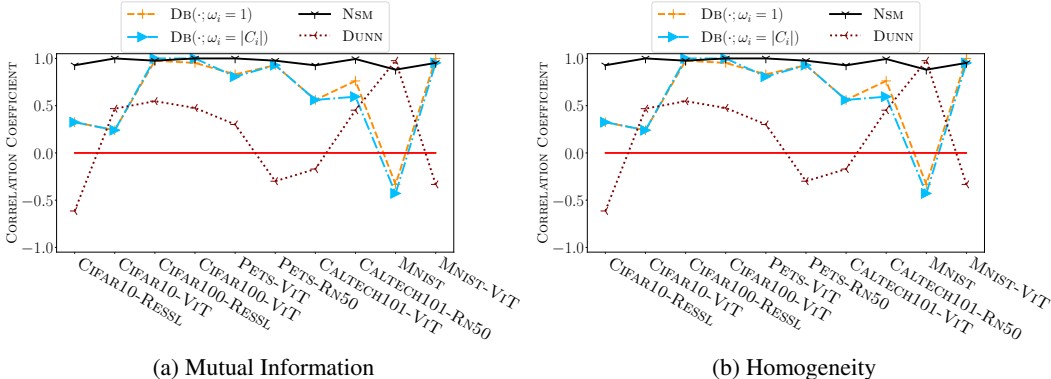

(a) Mutual Information             (b) Homogeneity

Figure 9: Spearman's correlation coefficient between clustering quality and external evaluation metrics for the task of image clustering. For DB, the coefficient is negated because a smaller index indicates better clustering quality. For clustering-NSM, we use approximate nearest neighbors to compute each set's stability.

## F    CLUSTERING-NSM USING APPROXIMATE STABILITY

In Section 4, we measured the clustering-NSM of a clustering by considering *exact* nearest neighbors to determine the stability of a set, as per Definition 1. That operation, however, can prove expensive for large datasets as it has a time complexity of $\mathcal{O}(n^2)$ for a collection of $n$ points.

In this supplementary experiments, we instead use *approximate* nearest neighbor to compute stability measures. We use clustering-based ANN search for this purpose, where we partition a dataset into $4\sqrt{n}$ clusters, and to find the approximate nearest neighbor of each point, we search only $10$ closest clusters. This change dramatically reduces the search time complexity.

The results of this experiment are shown in Figure 8 for the ANN search task, and Figure 9 for the image clustering task. Comparing the results with Figures 1 and 2 respectively reveals no perceptible difference in the standing of clustering-NSM relative to other measures.

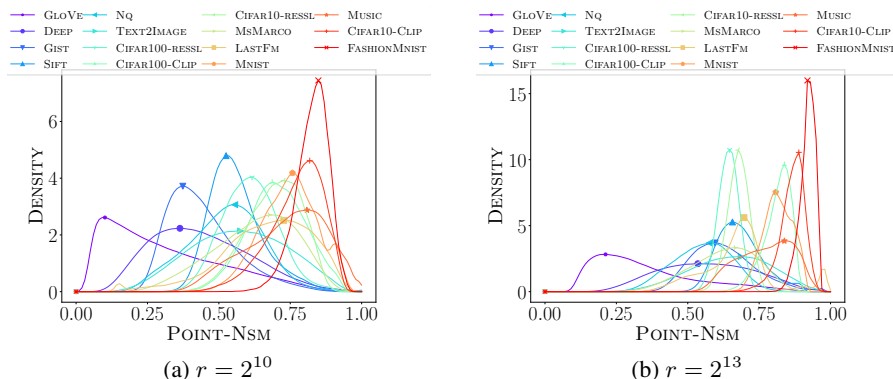

(a) $r = 2^{10}$                    (b) $r = 2^{13}$

Figure 10: Point-NSM distributions computed from all points within the various datasets.

Table 2: Spearman's correlation coefficient between statistics from point-NSM distributions (computed from full datasets) and the clustering-NSM for clusterings obtained from standard and spherical KMEANS. All correlations are significant ($p$-value $< 0.001$). Last row aggregates statistics and clustering-NSM for all values of $r$. These coefficients are exactly the same as those reported in Table 1.

| | STANDARD KMEANS | | SPHERICAL KMEANS | |
|---|---|---|---|---|
| POINT-NSM RADIUS | MEAN | 0.1-QUANTILE | MEAN | 0.1-QUANTILE |
| $r = 1024$ | 0.81 | 0.86 | 0.79 | 0.85 |
| $r = 2048$ | 0.82 | 0.85 | 0.82 | 0.85 |
| $r = 4096$ | 0.74 | 0.81 | 0.74 | 0.81 |
| $r = 8192$ | 0.76 | 0.78 | 0.76 | 0.78 |
| COMBINED | 0.79 | 0.83 | 0.82 | 0.85 |

## G   DISTRIBUTION OF POINT-NSM OVER THE FULL DATASET

The results presented in Figure 3 and Table 1 were based on the point-NSM distribution of $5\%$ of points in each datasets. In this section, we repeat the same experiments as in Section 5 but using the full datasets.

Figure 10 and Table 2, which parallel the figure and table in Section 5, show that the findings do not change—the correlation coefficients are exactly the same. This justifies the use a small sample to compute point-NSM, enabling high efficiency without any detriment to effectiveness.

