# OpenReview forum: "Neighborhood Stability as a Measure of Nearest Neighbor Searchability"
_ICLR.cc/2026/Conference — Submitted to ICLR 2026_

### Official Review · Reviewer_xvr7 · 2025-10-29

**Soundness:** 2
**Presentation:** 3
**Contribution:** 2
**Rating:** 2
**Confidence:** 4

**Summary:**

The task that the article considers is measuring the clusterability of the data set and a quality of a particular clustering of a data set for the purpose of measuring the suitability of the data set for the clustering-based approximate nearest neighbor search (ANNS). In particular, clustering-NSM (Neighborhood Stability Measure) is proposed for measuring the internal quality of a clustering. Clustering-NSM of a clustering is the average of set-NSM of the clusters, and the set-NSM of a cluster is simply the fraction of a cluster points whose nearest neighbor belongs to the cluster. The article also proposes a related clusterability measure that measures how likely a set of points is to have a high quality clustering by any clustering algorithm.

The experimental results show that the proposed quality measure (clustering-NSM) has a higher correlation with the accuracy of the ANN algorithms than the existing internal quality measures of clustering. In addition, the experimental results show that the proposed clusterability measure of a data set correlates with the quality (as measured by clustering-NSM) of the clustering that can be obtained on that data set.

**Strengths:**

The proposed quality measure of clustering has a higher correlation with the accuracy of the ANNS methods than the earlier quality measures of clustering. The proposed quality of clustering and clusterability measures are straightforward and intuitive. The article is well-written, and the formalization of the concepts and the mathematical notation are very good.

**Weaknesses:**

The proposed measures seem interesting for assessing the quality of clustering and the clusterability of a data set. However, the claim that the these measures enables determining whether a data set is searchable by clustering-based ANNS methods is insufficiently validated by empirical evidence. In particular:

- The experiments do not measure true end-to-end effectiveness of ANNS methods. In practice, the number of searched clusters $\ell$ is large, whereas in the experiments of the article $\ell = 1$, and SOTA clustering-base methods further prune the cluster points before exact distance computation, for instance by using product quantization (Jegou et al., 2010) or anisotropic quantization (Guo et al., 2020). Basically, the experiments are equal to doing a $k$-means clustering, and checking whether the nearest neighbors of the query point belong to the same cluster with it. Thus, while the result of the current experiment are promising, it is only a starting point, as it is not yet clear what is the effect of the clusterability of the data set or the quality of the clustering to the end-to-end effectiveness of SOTA clustering-based ANNS methods. See for instance, Aumüller et al, (2021) for an example of experimental design where the effect of different hardness measures of a data set to the effectiveness of ANNS methods are quantified (SOTA clustering-based methods are used, hyperparameter sweeps are performed, and whole recall-QPS curves at the optimal hyperparameters are reported).

- It is stated that the proposed quality measure of clustering enables comparing different clusterings of a data set to assess their effect on the performance of ANNS methods. However, the experiments do not test different clustering methods on a same data set to see whether the clustering-NSM of a clustering method is indicative of its end-to-end ANNS performance.

- It is claimed that the proposed clusterability measure enables determining the suitability of clustering-based methods for ANNS. However, no other types of ANNS methods are currently included in the empirical comparison. Thus, it is not clear what is the effect of the clusterability of the data set to the relative (compared to other types of ANNS methods, especially graph methods) performance of the clustering-based methods: if the data set has low clusterability, does it mean that clustering-based methods perform poorly on this data set compared to graph methods, or do all ANNS methods perform equally poorly on this data set?

- The proposed clusterability measure for assessing the suitability of the data set for clustering-based ANNS methods is only compared to the generic quality measures of a clustering. However, there already exists measures, such as relative contrast (RC) (He et al., 2012), local intrinsic dimensionality (LID) (Houle et al., 2013), and query expansion (Ahle et al., 2017) for assessing the hardness of a data set for ANNS; see, e.g.,  Aumüller et al, (2021) for empirical evaluation of these hardness measures.

In summary, while proposed measures seem interesting, the current empirical evaluation is only a brief "proof of concept"-study. As is, the manuscript would be a good workshop article, but acceptance to a top tier-conference would require much more thorough set of experiments to justify its claims.

References:

Ahle, Thomas D., Martin Aumüller, and Rasmus Pagh. "Parameter-free locality sensitive hashing for spherical range reporting." Proceedings of the Twenty-Eighth Annual ACM-SIAM Symposium on Discrete Algorithms. Society for Industrial and Applied Mathematics, 2017.

Guo, Ruiqi, et al. "Accelerating large-scale inference with anisotropic vector quantization." International Conference on Machine Learning. PMLR, 2020.

He, Junfeng, Sanjiv Kumar, and Shih-Fu Chang. "On the difficulty of nearest neighbor search." Proceedings of the 29th International Coference on International Conference on Machine Learning. 2012.

Houle, Michael E. "Dimensionality, discriminability, density and distance distributions." 2013 IEEE 13th International Conference on Data Mining Workshops. IEEE, 2013.

Jegou, Herve, Matthijs Douze, and Cordelia Schmid. "Product quantization for nearest neighbor search." IEEE transactions on pattern analysis and machine intelligence 33.1 (2010): 117-128.

**Questions:**

I do not have any questions for the authors.

---

> ### Author Response · Authors · 2025-11-15
>
> Thank you for reviewing our submission and for your feedback! Below you will find our response. We'd be delighted to engage in further discussion if you find our note unsatisfactory, but just so that the discussion continues in good faith, perhaps we could set aside opinions on whether our work is at best a workshop article or worthy of publication. Thank you!
>
> > The experiments do not measure true end-to-end effectiveness of ANNS methods....
>
> We measure how far a clustered dataset is from the ideally-clustered dataset wherein the top cluster contains the nearest neighbor. One can relax that criterion by letting $\ell > 1$, however one must also note that as $\ell$ increases recall/accuracy inevitably increases. When $\ell$ is sufficiently large, ANN search becomes almost perfectly accurate, and the difference in performance between clustering algorithms diminishes.
>
> That is the reason we chose to work with $\ell=1$. We would be happy to provide experimental results with $\ell>1$ if you clarify what you wish to understand from such an experiment that is not answered by the current draft.
>
> As an aside, we note that how scores are computed (exact, PQ, etc.) is irrelevant to the research question we ask in our work. Introducing approximation to score computation (such as by using quantization) can degrade ANN search accuracy independently of clustering quality or routing accuracy.
>
> Similarly, we fail to see how QPS is relevant to what we study in our work.
>
> > ... However, the experiments do not test different clustering methods on a same data set ...
>
> We do. See Section 4.1 for details.
>
> > ... However, no other types of ANNS methods are currently included in the empirical comparison....
>
> There are often questions around why IVF does not perform as well on one dataset as it does on other datasets (e.g., GloVe versus Music). Our method is an attempt to explain that empirical observation.
>
> We make no claims on whether other ANNS methods are appropriate for a dataset as that’s beyond the scope of our work. Instead, if one chooses a reference point (on the spectrum, say, from GloVe to Music), then Point-NSM can tell us how amenable a new dataset would be to clustering-based ANNS relative to the reference point.
>
> > ... However, there already exists measures, such as relative contrast...
>
> The measures you have cited are not measures of clustering quality, as such they are not comparable with Clustering-NSM *per se*.
>
> One could, however, use those measures to assess the searchability of a dataset. But all require nonnegative distance functions and cannot be applied to datasets where ANNS is based on inner product or cosine similarity.
>
> Our goal was to present a complete end-to-end set of measures: Point-NSM → Clustering-NSM → ANNS accuracy, where one is predictive of the next and where the measure does not require nonnegativity.

---

> ### Comment · Reviewer_xvr7 · 2025-11-17
>
> My point about measuring QPS was that, as you also state in your reply, accuracy alone is insufficient to assess the quality of an ANN algorithm: if you reduce the number of clusters, you will improve the accuracy, and with only one cluster consisting of  all the data points you of course get a perfect accuracy. Thus, it is necessary to consider both _accuracy_ (typically measured by recall) and _efficiency_ (typically measured by query time or, equivalently, by QPS).
>
> I think it is a relevant and interesting research question to assess the relative performance of clustering-based ANN algorithms compared to other types of ANN methods, especially graph methods, and that is why I suggested it. And to that end, you also need to measure QPS in addition to accuracy. From your introduction I got an impression that you were tackling this more ambitious research question.
>
> But if you decide to frame your research question narrowly, as you do in your reply, as measuring only the accuracy of clustering-based ANN methods, then I have to agree with Reviewer iRDX that the proposed measure is a bit tautological: if you define your measure as the fraction of the points whose nearest neighbor belongs to the same cluster with them, and then define the external quality measure measure as a proportion of the nearest neighbors of the point that belong to the same cluster with it (when you measure ANN accuracy with $\ell = 1$ and no PQ, it comes down to this), then it is not a very surprising or insightful result that these two metrics are highly correlated.

---

> > ### Author Response · Authors · 2025-11-17
> >
> > **On the scope of our research question**: The research question we set out to investigate is, at a high level, to understand why clustering-based ANN search performs "better" on some datasets than others (see e.g. [1]). For example, from [1], it is clear that search using IVF on Glove-100 generally leads to a recall-qps curve that is below that of Fashion-MNIST's.
> >
> > There is currently no theoretical explanation for the phenomenon. As we also discuss in the submission, existing statistical tools either can't be meaningfully applied to datasets with possibly negative distance/similarity measures, or they do not correlate with ANN search performance.
> >
> > What we present is the first tool to make sense of those empirical observations. Evidence suggests that the distribution of point-NSMs appears to explain the relative performance of clustering-based ANN search (IVF) across datasets. For example, the distribution of Fashion-MNIST is concentrated closer to 1 whereas Glove (albeit here we use Glove-200) is closer to 0.
> >
> > This method is practical too: by sampling a small subset of a dataset and forming the point-NSM distribution, one can set expectations as to whether IVF leads to reasonable performance. This is scalable, can be applied to even inner-product based datasets, and gives insight into expected IVF performance independent of the clustering algorithm.
> >
> > **On QPS, PQ, and other efficiency-focused experiments**: We are not entirely confident we understand the relevance of measuring QPS (or quantizing points for faster score computation) to our research question. A simple fact you may want to consider is that, to reach a fixed level of ANN recall (say, 90%), IVF would have to "probe" more clusters if the clustering of the data is poor, and fewer clusters if the clustering is of higher quality. Probing more clusters translates to a higher latency, because latency is additive (assuming a single CPU thread).
> >
> > **On IVF vs graph vs other ANNS methods**: That is indeed a great question, but one that is beyond the scope of our research and one that has been studied extensively. You can refer to existing publications for a comprehensive comparison (e.g., [2]).
> >
> > **On comment implying tautology**: As we noted in our response to Reviewer iRDX, it is not the case that queries are sampled from the same distribution as data points.
> >
> > Like we suggested, the method helps make sense of a phenomenon that is well-documented in the literature but remains unexplained (until our study). It is insightful in that sense, and it does benefit the research community as well as practitioners.
> >
> >
> > --------
> > [1] https://ann-benchmarks.com/faiss-ivf.html
> > [2] https://arxiv.org/pdf/2401.08281

---

> > > ### Author Response · Authors · 2025-11-17
> > >
> > > We should add one more detail that may have escaped your attention: In Section 4.1, the number of clusters is *fixed* as we change the clustering algorithm on a single dataset. Clusters are typically balanced in size, so that probing the same number of clusters leads to similar score computation latency. So a situation you noted, where we may have a single large cluster, is a variable that's being controlled.

---

> ### Author Response · Authors · 2025-11-19
>
> On reflection, we think QPS was a red herring that distracted us from the deeper question you may have intended to raise: Do these clustering quality measures correlate with ANN search accuracy as we probe more and more clusters ($\ell$)? One can think of $\ell$ as a proxy for efficiency.
>
> ​We have just updated our submission with Appendix E (pages 17 and 18) that presents experimental results on all ANN datasets to answer that question. You will find one figure per dataset where we plotted the recall-$\ell$ curves for reference. Additionally, we repeated the experiment which led to Figure 1 for every value of $\ell$.
>
> ​As we explain in the appendix, there are two trends to highlight: a) the performance of clustering-NSM relative to other measures remains stable across values of $\ell$, with one exception; and b) all correlation coefficients degrade on most datasets as $\ell$ becomes larger, which is to be expected as ANN accuracies from different indices get closer.
>
> ​Does this additional result address your original concern?

---

> > ### Comment · Reviewer_xvr7 · 2025-11-21
> >
> > Thanks for taking into account my feedback and updating the experimental results.
> >
> > However, my main concern still remains: as you state in your response, you limit your research question to studying why clustering-based ANN methods perform better on some data sets than others. While you answer that research question in your article, I think this is a too narrow question to have practically significant effect even it is solved.
> >
> > Right now, one cannot know whether clustering-based methods perform worse on glove-100 than fashion-MNIST because glove-100 is a hard data sets _specifically_ for clustering-based methods (compared to other types of ANN methods), or is it the case that glove-100 is a harder data set than fashion-MNIST for ANN search in general. To show that your proposed measures have some explanatory power, you should measure their effect also to other types of ANN methods, and compare the proposed measures to the other existing hardness measures for ANNS (mentioned in my original review) to find out if there is correlation with them, or if the proposed measure is orthogonal (Yes I know that the clustering measure can be used only for clustering-based methods, but the proposed clusterability measure can be more widely applied as a hardness measure).
> >
> > In addition, I would like to point out that on the most of the data sets (with the exception of the information retrieval data sets, such as MS-Marco and T2I), the queries and the database are indeed drawn from the same distribution.

---

> ### Author Response · Authors · 2025-11-21
>
> Thank you for considering our additional experiments!
>
> Just to summarize the discussion so far: Our understanding is that you agree that our research question as stated in the work is satisfactorily answered by the results and experiments we provided: That our clustering quality measure can inform us on the performance of ANNS, and that the searchability measure is predictive of clustering quality measure, thereby creating an end-to-end statistical tool for clustering-based ANNS (searchability predicts clustering quality which predicts ANNS performance). But that you (a) question the practical utility of our research and (b) believe an expansion of the scope to other ANNS methods would enhance our work.
>
> (a) **Utility**: As we mentioned in our draft, there are several vector databases that have based their technology on clustering-based ANNS (we have provided example links), many of them serving as retrieval engines for other upstream tasks to which retrieval latency is critical. They take an often information retrieval dataset, cluster it, and, to lower costs, place "shards" on high-latency storage. Shards are then fetched from storage during query processing and searched. This procedure is detailed in the blog posts we linked, but there are research articles that explore problems within this design.
>
> One way our method can be used in that context is resource allocation such as deciding which datasets' shards need to rest in the query processor's cache with a higher priority. Datasets that are "harder" according to point-NSM often need more of their shards searched for the vast majority of the query distribution, so that it makes sense to allocate more cache for them, relative to easier datasets who only needs a large number of their shards for tail queries.
>
> As another, it can be used to decide if the indexing should increase the number of clusters for a dataset and "pay" more in query routing for the sake of higher accuracy with a smaller $\ell$. That decision can be guided by the distribution of point-NSM.
>
> Yet another is by monitoring a dynamic dataset as one stores new vectors or removes existing ones. As the point-NSM distribution changes, one can adjust indexing parameters.
>
> We hope these examples give you a sense of the practicality of our method.
>
> (b) **Extension to other ANNS algorithms**: We do not disagree that studying point-NSM in the context of other ANNS methods (in particular, graph methods) is a worthwhile research direction. We even noted that in our concluding section. But we also believe this is a significant enough effort that warrants its own study. We again note that none of the existing methods is capable of handling possibly negative similarity functions (such as inner product).
>
>
> -----
>
> We are grateful for the discussion here and for the healthy skepticism. We hope we were able to alleviate some of your concerns. If you believe we should adjust any of our claims, or add a note to the draft clarifying the applicability of our method, or present new experiments, we'd be happy to do so.

---

> > ### Author Response · Authors · 2025-11-26
> >
> > Hi! Since we're getting close to the end of the discussion period, we wanted to reach out to see if you've had a chance to consider our note above. Are there still issues we should discuss and concerns we may be able to address?
> >
> > Our hope is to show that our work is not only technically correct but that it also has utility in real-world applications. While we understand an extension of this work to other ANNS algorithms would enhance the work, we still believe the findings documented in our current submission are worth sharing with the community.

---

> > > ### Comment · Reviewer_xvr7 · 2025-11-27
> > >
> > > Thanks for your thorough responses. However, I maintain score since my view is still that while the work is technically correct, the contribution of the current version of the manuscript is not significant enough to warrant publication.

---

> > > > ### Author Response · Authors · 2025-12-01
> > > >
> > > > While we appreciate the discussion, we must admit it's difficult to present an argument when the goalposts shift from comment to comment. To summarize the discussion thus far, we have:
> > > >
> > > > * presented additional experiments with $\ell > 1$
> > > > * explained why existing measures are not applicable (they cannot work with possibly negative similarity functions such as inner product)
> > > > * articulated why the findings are not trivial: query points can be drawn from a distribution distinct from data points, as in many large-scale information retrieval datasets
> > > > * gave concrete examples of why our method is useful in practice
> > > > * elaborated our focus on clustering-based ANN search--a widespread paradigm--and stated our intention to study other ANN search methods in future (our theory at the moment only justifies application of the method to clustering-based ANN search).
> > > >
> > > > As to why our contribution is "significant enough to warrant publication": As we clearly state in the submission, our proposal is the first principled statistical method that explains why clustering-based ANN search works well on some datasets but not on others. Principled methods are hard to come by in the modern ANN search literature, especially if they can be easily applied. Our proposal is both principled and applicable. We also explain in what ways the method can be applied.
> > > >
> > > > We understand the discussion period was cut short due to a security incident. But we wanted to present this summary and our final arguments for posterity.

---

### Official Review · Reviewer_epcJ · 2025-10-31

**Soundness:** 2
**Presentation:** 3
**Contribution:** 3
**Rating:** 6
**Confidence:** 4

**Summary:**

This paper proposes an internal measure (Clustering-NSM) for measuring clustering quality and a dataset-dependent measure (Point-NSM) for measuring the clusterability of a dataset. The experiments demonstrates that Clustering-NSM is highly correlated with ANNS accuracy and external measures for clustering quality. In addition, the paper proves that a high and concentrated Point-NSM indicates better Clustering-NSM and validates this theorem with a clustering task on image datasets.

**Strengths:**

-	It is very novel that Theorem 1 bridges clustering-dependent measure (Clustering-NSM) and clustering-independent measure Point-NSM. As a measure that is only dependent on the dataset, Point-NSM provides valuable insights on the clusterability of the dataset and following clustering and ANNS task.
-	Both Clustering-NSM and Point-NSM are independent of the choice of distance metric. This makes both the measure have boarder flexability.
-	The experiments are conducted on a range of different types of datasets and many types of vision model for encoding. The results are convincing.
-	The design that different iteration steps of k-means indicates different clustering quality is interesting and justified.

**Weaknesses:**

-	The high correlation between internal measure such as Clustering-NSM and external measure is surprising. However, this result is only empirical. Is there any explanation for this result?
-	The choice of external measure is very limited. Other popular measures such as (Adjusted) Rand Index could be used.
-	A minor point. Don’t focus on this if the author do not have time for this issue. Can the proposed Clustering-NSM be used for hyperparameter search? It would be interesting to use NSM in unsupervised task like clustering to determine hyperparameter. Many methods now just use the ground-truth label, i.e. external measure like NMI, for hyperparameter search. This is highly impractical and unjustified based on the unsupervised nature of the task. A method with more hyperparameters or continuous hyperparameter often gives a very good yet unfair results through a grid search with very small steps. It would be interseting to see the use of NSM on this aspect.

**Questions:**

See Weaknesses.

---

> ### Author Response · Authors · 2025-11-13
>
> Thank you for taking the time to examine our submission and for recognizing the strengths of our proposal in your thoughtful review! Below is our response to your questions.
>
> > The high correlation ... is surprising. However, this result is only empirical. Is there any explanation for this result?
>
> If we may offer an intuitive explanation: Suppose $X$ is a dataset with a non-trivial clustering whose Clustering-NMS is $1$. It is easy to see that each cluster is concentrated around their mean *relative* to its distance to other clusters. So that if a query point is closer to that cluster's centroid, then it is also likely that its nearest neighbor also lies in the same cluster.
>
> If, on the other hand, a clustering of $X$ results in Clustering-NSM of $0$, then clusters are more "spread out" in a sense. So that queries are more likely to be near the boundary of multiple clusters, making routing more error prone.
>
> It is an interesting question as to whether this intuition can be formalized. But unfortunately making formal statements on the accuracy of ANN search is a nontrivial research question: Among graph- and clustering-based methods, there is only one formal result on a theoretical algorithm only [1]
>
> > The choice of external measure is very limited. Other popular measures such as (Adjusted) Rand Index could be used.
>
> Yes, that is regrettable. As we explained in the draft, however, it is not for lack of awareness or trying: Other indices/measures unfortunately do not scale to the dataset sizes that are commonplace in the ANN search literature -- in fact, scalability is one strength of our proposal.
>
> As for (adjusted) Rand Index in particular, we note that it is an external measure of clustering validity, requiring ground-truth cluster assignments.
>
> > Can the proposed Clustering-NSM be used for hyperparameter search?
>
> That is indeed a great suggestion and an exciting idea to explore!
>
> [1] Indyk and Xu. "Worst-case Performance of Popular Approximate Nearest Neighbor Search Implementations: Guarantees and Limitations." NeurIPS 2023.

---

> > ### Comment · Reviewer_epcJ · 2025-11-25
> >
> > Thank you for the responses. I have no furthur questions. The respones have address my concerns and I will maintain my score.

---

> > > ### Author Response · Authors · 2025-11-26
> > >
> > > Thank you for taking our note into consideration! Are there other concerns that you may have that make our work marginally acceptable, rather than acceptable? We'd be happy to either try to address them or in a future revision.

---

### Official Review · Reviewer_h922 · 2025-10-31

**Soundness:** 3
**Presentation:** 3
**Contribution:** 3
**Rating:** 4
**Confidence:** 3

**Summary:**

This paper introduces two measures for evaluating the suitability of clustering-based Approximate Nearest Neighbor Search (ANNS). The clustering-NSM measures the proportion of points in a cluster whose nearest neighbors also lie in the same cluster, while the point-NSM measures, for each point, how many of its nearest neighbors share stable neighborhood relationships. The clustering-NSM is shown to be strongly predictive of point-NSM.
These measures generalize k-nearest neighbor consistency to a continuous score between 0 and 1. The clustering-NSM serves as an internal clustering quality metric, and the point-NSM reflects dataset clusterability under flat clustering.
Extensive experiments show that clustering-NSM correlates strongly with external metrics such as ANN accuracy, mutual information gain, and outperforms traditional indices like Dunn and Davies–Bouldin. The results also demonstrate that clustering-NSM can predict point-NSM, enabling dataset searchability assessment without requiring query distributions—an important benefit for large-scale search systems. Furthermore, the experiments indicate that point-NSM can serve as a measure of the dataset’s overall clustering quality.

**Strengths:**

(S1) The paper proposes a theoretically sound and novel clustering measure that fulfills the four axioms of Ben-David and Ackerman (2008), which are considered fundamental requirements for a clustering quality function. It establishes a theoretical framework showing that clustering-NSM satisfies these axioms. Furthermore, the authors derive probabilistic bounds connecting point-NSM and clustering-NSM under the assumption that the data follow a spherical flat clustering structure and are uniformly distributed.

(S2) The paper conducts extensive experiments on benchmark datasets to evaluate the suitability of the proposed measures for k-nearest neighbor searchability analysis and to assess their usefulness in quantifying dataset clusterability. The results demonstrate a strong correlation between clustering-NSM and top-k approximate nearest neighbor accuracies, outperforming the Dunn index and the Davies–Bouldin index. The experimental evidence is convincing.

(S3) The paper is well written, with a clear presentation of the theoretical contributions, logically inherent, and comprehensive empirical validation.

**Weaknesses:**

(W1) Generalizability of the empirical evaluation of point-NSM for cluster ability:
The computation of point-NSM requires nearest-neighbor calculations across the entire dataset, which can be computationally expensive. The authors acknowledge this in their empirical evaluation of point-NSM for clusterability, where they mitigate the cost by randomly subsampling 5% of the points to estimate the point-NSM distribution. But does the contribution stay the same when increasing the amount of subsampling to 100%?  Would we still observe this mostly positive correlation between point-NSM and cluster-NSM?

(W2) Theoretical Richness:
The interaction between the hyperparameter ω (the weighting of cluster-NSM) and the hyperparameter 𝑟 (the neighborhood size used for point-NSM) is not thoroughly analyzed. This may help explain why, as shown in Figure 3, not all datasets exhibit a clear positive correlation between clustering-NSM and point-NSM.

(W3) Restricted theoretical scope:
In the state-of-the-art review, the paper does not reference some recent advances in clusterability research (e.g., https://arxiv.org/pdf/1808.08317 or https://arxiv.org/pdf/2310.12806). Including discussions of these works would provide better context and help position the proposed contributions more clearly. Similarly, in the experimental evaluation, it would be valuable to compare clustering-NSM with more recently developed clustering quality metrics beyond the Dunn and Davies–Bouldin indices. The analysis focuses exclusively on flat clustering algorithms (e.g., K-means and spherical K-means). The exclusion of hierarchical, graph-based, or density-based methods (e.g., DBSCAN or spectral clustering) limits the generality of the conclusions. This raises the question of whether point-NSM can truly quantify the difficulty of clustering a dataset.

**Questions:**

(Q1) In Figure 3, we observe that the correlation between point-NSM and clustering-NSM varies across datasets, and in some cases (e.g., GloVe), the correlation appears to be weak. How would you revise the paper to explain why this occurs?

---

> ### Author Response · Authors · 2025-11-13
>
> Thank you for reviewing our work and for your thoughtful feedback! We appreciate your recognition of the strengths of our work as well. Below is our response to your questions and comments.
>
> > ...But does the contribution stay the same when increasing the amount of subsampling to 100%?
>
> We're assuming data points are sampled iid from an unknown distribution $\mathcal{D}$---a common assumption. The distribution of point-NSMs of a set $X \sim \mathcal{D}$, where $\lvert X \rvert$ is large enough, should adequately reflect the distribution of poin-NSMs of $\mathcal{D}$. That's effectively what we have done.
>
> While we do not present results in our current draft, in limited experiments, we notice no meaningful change (either to the shape of the point-NSM distributions or the computed correlations) by increasing the sample size. That makes sense, as we explained above, large-enough samples are representative of the underlying data distribution.
>
> If you believe it is important to demonstrate this empirically, we'd be happy to update the draft with further experiments.
>
> > The interaction between the hyperparameter ω (the weighting of cluster-NSM) and the hyperparameter 𝑟 (the neighborhood size used for point-NSM)
>
> We agree that it would be interesting to study the theoretical and empirical effects of $\omega$. As we noted in the submission, we'd like to study this separately in the future. The reason being that $\omega$ plays a major role in the theoretical results (see Theorem 2), potentially introducing complexity that deserves its own thorough investigation.
>
> > In the state-of-the-art review, the paper does not reference some recent advances in clusterability research
>
> We do indeed cite Adolfsson et al.'s work on clusterability. The second work you referenced is on *external* measures of clustering quality, which is not related to our proposal.
>
> We'd like to further note that, as we stated in our response to Reviewer iRDX, what we have proposed is not interchangeable with general clusterability: Any dataset can be made more searchable: by increasing the number of clusters, ANN search accuracy improves. While the traditional notion of clusterability is agnostic to the number of clusters, searchability (our definition) is a function of cluster size---a proxy for the number of clusters.
>
> > it would be valuable to compare clustering-NSM with more recently developed clustering quality metrics beyond the Dunn and Davies–Bouldin indices
>
> As we explained in the draft, that is not for lack of awareness or trying: Other indices/measures do not unfortunately scale to the dataset sizes that are commonplace in the ANN search literature -- in fact, scalability is one strength of our proposal.
>
> > The analysis focuses exclusively on flat clustering algorithms...
>
> That is part scalability part practicality. Some clustering algorithms do not efficiently scale to ANN datasets. But more importantly, it is not clear how one should route queries to the right clusters: Given a hierarchical or spectral clustering of a dataset, how should clusters be represented (centroids? core-sets?) so that we may identify the top clusters for a query. That is a research question that is beyond the scope of our work.
>
> > In Figure 3, we observe that the correlation between point-NSM and clustering-NSM varies across datasets...
>
> We're not entirely confident we understand the question. Figure 3 shows the distribution of point-NSM for various datasets using different values of $r$. Table 1 summarizes the correlation between clustering- and point-NSM statistics, but it does not present statistics on individual datasets.

---

> > ### Comment · Reviewer_h922 · 2025-11-21
> >
> > The paper uses the idea of k-nearest neighbor consistency to quantify clusterability. The parameter k is a hyperparameter and influences the measurement of clusterability. If the data are distributed with varying densities, then for a fixed k, it is likely that points in sparse regions will include neighbors from dense regions. As a result, the NSM score may be lower. However, this does not necessarily mean that the dataset is difficult to cluster.
> >
> > Under the given i.i.d. assumption, the scenario described above will not occur. Nevertheless, it remains difficult for us to assess the impact of your contribution, and we cannot conclude that a low cluster-NSM score implies that partitioning-based clustering methods, such as k-means, will perform poorly on the dataset.
> >
> > Maybe a discrimination from "Fast k-Nearest-Neighbor-Consistent Clustering" by Lenssen, Strahmann, Schubert at LWDA 2023 (https://ceur-ws.org/Vol-3630/LWDA2023-paper34.pdf) helps clarify the novelty.
> >
> > Therefore, thank you for your explanations, we appreciate that but keep our grading.

---

> ### Author Response · Authors · 2025-11-21
>
> Thanks for your note! Is your objection regarding the sampling (5%) we performed to arrive at the point-NSM distribution and Table 1? Would your concern be addressed if increasing the sample size to 100% corroborated our statement above?

---

> > ### Author Response · Authors · 2025-11-22
> >
> > Dear Reviewer h922 - we have repeated the experiments of Section 5 on the full dataset (i.e., computing point-NSM distributions on 100% of points in each dataset). As Appendix G shows in the updated draft, the results do not change: While the shape of each distribution changes very subtly (see Figure 3 vs. Figure 10), the correlation coefficients reported in Table 2 in the appendix match measurements in Table 1 exactly.
> >
> > We hope this addresses your main concern, but we're happy to answer any further questions you may have.

---

> > > ### Author Response · Authors · 2025-11-26
> > >
> > > We hope you've had a chance to consider our additional experiments. Are there other concerns or issues we should discuss or attempt to address?

---

### Official Review · Reviewer_iRDX · 2025-11-01

**Soundness:** 3
**Presentation:** 3
**Contribution:** 2
**Rating:** 2
**Confidence:** 4

**Summary:**

This paper addresses an important and practical problem: determining the "searchability" of a dataset for clustering-based Approximate Nearest Neighbor (ANN) search. The main contribution is the introduction of the "Clustering-NSM" measure, which the authors claim can  predict the performance of the dataset on a clustering based ANN data structure.

**Strengths:**

Picking an appropriate ANN datastructure for a dataset is a very practical problem as there are many choices available to practitioners. Furthermore, the introduced measure is demonstrated to have a high correlation with performance on (simple) ANN datastructures based on clustering.

**Weaknesses:**

- One of the paper's drawback is the following: it attempts to define and measure a dataset's suitability for clustering-based ANNS, which is a function of a specific clustering algorithm, rather than simply measuring the dataset's clusterability itself. Thus, it is not clear to me why one would study "searchability" (a function of a clustering) instead of the dataset's intrinsic clusterability. In what practical situation would a dataset that is "clusterable" (e.g., as measured by k-means loss) not be "searchable" by a clustering-based ANNS algorithm? I am not sure if this distinction exist.

- The introduced measure is also not very surprising as it seems to directly mimic testing the ANN datastructure on 'queries', where the performance on queries is approximated by performance on the dataset points themselves (To me this is what is going on since query points and the input dataset points are often from the "same distribution." The proposed Clustering-NSM measure is defined as the fraction of points whose 1-nearest-neighbor resides in the same cluster. For a clustering-based ANNS method, high recall typically requires that a query's true nearest neighbor be in the same cluster. Again, assuming query points are drawn from the same distribution as the dataset, Clustering-NSM is not just predictive of ANNS accuracy, but rather seems like direct proxy for it. So it is not clear to me if the high correlation shown is particularly insightful.

- Another major practical drawback is the computational cost of the proposed measures. To compute Clustering-NSM one must find the nearest neighbor(s) for every point in the dataset (ignoring the point itself). This is a computationally intensive task, requiring quadratic time in the worst case. This is a high cost and a significant barrier in practice. The authors claim that one can replace this by finding approximate nearest neighbors, but this seems a bit circular to me: then we need ANN data structure to compute this measure but if we had the data structure already, then one could simply test the performance of the ANN datastructure directly on the dataset, making the measure useless.

- I am also not convinced by how interesting Theorem 1 is. It uses an imprecise and subjective statement "As such, clustering-NSM is a measure of clustering quality." This statement seems to be in reference to a set of very specific axioms presented in an old paper by Ben-David & Ackerman (2008). However, it provides no justification for why this specific set of axioms should be considered the standard. Given that these axioms do not appear to be popular in the literature, the authors should justify why they are the "right" axioms to use.

- There also appears to be either an error  or a typo in the statement of Theorem 2. The right hand side of the inequality has a *minus* $\sqrt{\log(1/\epsilon)}$ term. If we let $\epsilon$ approach 0  the term $\log(1/\epsilon)$ approaches positive infinity. This causes the right-hand side of the inequality to approach negative infinity. A measure like clustering-NSM, which is a fraction and thus non-negative, cannot be less than a value approaching negative infinity.

- On a minor note, the paper consistently refers to "inner product distances". This is mathematically imprecise. An inner product is a measure of similarity, not a distance, as it does not satisfy the axioms of a metric (e.g., non-negativity). "Similarity" would be the more appropriate term.

**Questions:**

Do the authors have examples of datasets where the dataset itself is not a good candidate for clustering (i.e. it has poor "clusterability") but performs well for clustering based ANN datastructures or vice versa? Such examples (or understanding why such examples can arise) would convince me more of why one should study which datasets are "good" for clustering based ANN datastructures, rather than studying the simpler question of which datasets are "good" for clustering itself. Right now, it seems that the first question is a function of the answer of the second question.

---

> ### Author Response · Authors · 2025-11-13
>
> Thank you for taking the time to review our submission and for sharing your feedback! Below is our response.
>
> > ... In what practical situation would a dataset that is "clusterable" (e.g., as measured by k-means loss) not be "searchable" by a clustering-based ANNS algorithm? I am not sure if this distinction exist.
>
> A quick clarification: Clustering quality is a function of a *clustering* of a dataset. Clusterability is a measure of the dataset itself, independent of the clustering function. The k-means loss could be considered a measure of *clustering quality*, but not *clusterability*.
>
> Any dataset can be made more searchable: by increasing the number of clusters, ANN search accuracy improves. While the traditional notion of clusterability is agnostic to the number of clusters, searchability (our definition) is a function of cluster size---a proxy for the number of clusters.
>
> We recognize that, we regrettably did not make that distinction as clear as we should have in the current draft. We will address that in our final revision. But we're eager to learn if our answer clarifies the point and addresses your concern.
>
> >  The introduced measure is also not very surprising...
>
> It is not generally the case that query points are sampled from the data distribution. In our experiments, MS MARCO, Text2Image, and NQ are examples where queries have a different distribution than data points (e.g., text documents versus text queries).
>
> > Another major practical drawback is the computational cost of the proposed measures.
>
> Finding the nearest neighbors of the points in a dataset is an upfront and one-time cost. Once that is obtained, one can compute the clustering-NSM of any clustering algorithm.
>
> Computing the Point-NSM---which can tell us if clustering-based ANN search is even appropriate---requires the nearest neighbors of a small sample of the data distribution, leading to a time complexity of $\mathcal{O}(mn)$ for a dataset of size $n$ and a sample size of $m$ points.
>
> As for using approximate instead of exact nearest neighbors: One may use a graph-based index, or use an IVF (clustering-based) index but with a relatively large $\ell$ to obtain near-exact nearest neighbors. One can further utilize quantization, instruction-level parallelism, and a suite of other optimization techniques to speed up distance computation.
>
> > I am also not convinced by how interesting Theorem 1... the authors should justify why they are the "right" axioms to use.
>
> We must choose an axiomatic definition of what a clustering quality is; otherwise the notion would be imprecise. The Ben-David and Ackerman axioms constitute an established set of axioms in the literature. We are not in a position to argue what axioms are "right" for a measure of clustering quality, but would gladly consider axioms that you believe more appropriately define the expected behavior of a clustering quality measure.
>
> > A measure like clustering-NSM, which is a fraction and thus non-negative, cannot be less than a value approaching negative infinity.
>
> The theorem is correct as is. As $\epsilon \rightarrow 0$, the right-hand-side approaches $-\infty$. That inequality happens with probability at most $\epsilon$, which tends to $0$. That makes sense.
>
> > On a minor note, the paper consistently refers to "inner product distances"
>
> We settled on "distance" as it is more common in the ANN search literature, where inner product-based distance is typically understood as negative inner product similarity. But we agree with you in principle and will make an effort to state this more precisely in our final revision.
>
> > Do the authors have examples of datasets where the dataset itself is not a good candidate for clustering...
>
> Please see our answer to your first question. If that does not address this question, we're happy to discuss further.

---

> > ### Author Response · Authors · 2025-11-26
> >
> > We are writing to follow up on this thread to see if you've had a chance to read our note, and if you have further questions or concerns. Anything that we could discuss or attempt to address?

---

### Official Review · Reviewer_gKg9 · 2025-11-05

**Soundness:** 3
**Presentation:** 3
**Contribution:** 2
**Rating:** 4
**Confidence:** 3

**Summary:**

This paper identifies and addresses a practical gap in the field of clustering-based Approximate Nearest Neighbor Search (ANNS): the lack of analytical tools to determine if a dataset is "searchable," (i.e., well-suited for this search method) before running costly experiments. The authors propose a two-part solution: (1) Clustering-Neighborhood Stability Measure (clustering-NSM): An internal clustering quality measure that, unlike existing measures, is shown to be predictive of the external task-based measure of ANNS accuracy (recall). (2) Point-Neighborhood Stability Measure (point-NSM): A novel "clusterability" measure that operates on the dataset itself. This measure is predictive of the clustering-NSM.

The central claim is that this chain of prediction (point-NSM $\rightarrow$ clustering-NSM $\rightarrow$ ANNS accuracy) allows practitioners to assess a dataset's searchability given only the data points, a significant practical advantage. A key strength of these measures is their foundation on nearest-neighbor relationships rather than absolute distances, making them applicable across various distance/similarity functions, including Euclidean, cosine, and inner product. These are the first measures to serve as a tool for quantifying the amenability of a set of points to clustering-based ANNS.

**Strengths:**

Novel and High-Impact Problem: The problem of a priori algorithm selection is difficult and valuable. The paper's focus on "searchability" addresses a pain point familiar to any practitioner who has had to "guess and check" ANN indexing strategies.

Practical Utility: If the proposed measures are computationally efficient, they could save significant time and resources by providing a strong signal for or against using clustering-based ANN without the need to complete the costly experiment.

Generality of Measures: The authors' choice to base the measures on k-NN relationships (specifically 1-NN) instead of raw distances is a key insight. This makes the framework robust and applicable to inner product spaces, which are common in modern vector search (e.g., for embeddings) but are problematic for many traditional clustering measures that require non-negative distances.

Further, the authors provided experimental evidence of a potentially strong connection between proposed stability measure and ANN search results.

**Weaknesses:**

The major incentive for designing NSM's is to save the run-time of the whole clustering + ANN approach, hence the utility of point-NSM is entirely dependent on its computational complexity. The measure is described as a "statistic summarizing the distribution of point-NSMs," where a single point's NSM is derived from its "r nearest neighbors." Calculating nearest neighbors for all points can be quadratic operation, which is prohibitive. I think the point of clustering is to "shrink" the scope of NN searches for any given query point. It's unclear to me how much time we can save by directly computing the Point-NSMs. I found both the theoretical and empirical discussion to be dissatisfying to that end.

It's also curious that the benchmark metrics used in the experiments, DUNN and DB, were proposed in the 1970s. It seems unlikely that this is the STOA approach for either evaluating clustering + ANN method effectiveness, or "clusterability" of any dataset.

**Questions:**

Why is the definition of Point-NSM introduced half-way through the discussion on Set- and Cluster-NSM in section 3.2? It doesn't look like it is used anywhere in that section.

---

> ### Author Response · Authors · 2025-11-13
>
> Thank you for taking the time to review our submission! We appreciate your recognition of our work's strengths and your thoughtful questions. Below is our response to specific parts of your feedback.
>
> > Computational complexity
>
> You're correct to suggest that the complexity of identifying the exact nearest neighbors is $\mathcal{O}(n^2)$ for a dataset of size $n$. However:
>
> * to compute Clustering-NSM, as we show in Appendix E, it is sufficient to consider the *approximate* nearest neighbors, which can be computed in $\mathcal{O}(n \log n)$.
> * to compute Point-NSM, one only needs a small sample of size $m \ll n$ from the data distribution, rendering the exact search an $\mathcal{O}(mn)$ operation or approximate search $\mathcal{O}(m \log n)$.
>
> Finally, finding the nearest neighbors of the dataset is an upfront and one-time cost. Once that is obtained, one can compute the clustering-NSM of any clustering.
>
> > Clustering quality metrics
>
> The literature on clustering quality is not as fast moving, so naturally some well-established methods date back to the 70s. We do, however, explain why our choices were so limited: Other indices/measures unfortunately do not scale to the dataset sizes that are commonplace in the ANN search literature. In fact, one strength of our proposal is that it is easily scalable (with $\mathcal{O}(n \log n)$ complexity as discussed above).
>
> > Why is the definition of Point-NSM introduced half-way through the discussion on Set- and Cluster-NSM in section 3.2?
>
> We thought presenting the definitions in one section would make the draft more readable. But we are happy to re-organize if anything seems out of place.

---

> > ### Author Response · Authors · 2025-11-26
> >
> > We hope you've had a chance to consider our note above. Do you have further questions or concerns that we should discuss or attempt to address?

---

### Meta-Review · Area_Chair_xi93 · 2026-01-07

**Summary:**

The reviewers point out the following strengths of the paper:
- Clustering-NSM correlates with the performance of clustering-based ANNS. Thus, clustering-NSM can be used, e.g., to select a better clustering for NNS.
- Point-NSM is a measure of clusterability of the dataset; it correlates well with clustering-NSM. Thus, one can compare (clustering-based) searchability of datasets with this measure.
- Measures can be applied to arbitrary distance/similarity functions.

On the other hand, there are several limitations mentioned by the reviewers. While some of the reviewers' concerns have been addressed in the rebuttal, there are remaining limitations, and the reviewers’ ratings remain borderline after the discussion. One of the concerns is about the limited scope of the paper: while point-NSM allows one to compare the searchability of datasets, it does not allow one to decide which NNS method to apply, since no conclusions can be made about relative performance compared to other approaches. Also, to compute this measure, one actually needs to search for (approximate) nearest neighbors, and thus, some NNS index is needed at this step. Finally, the obtained correlation between clustering-NSM and NNS performance is expected since this measure, by construction, mimics the performance of clustering-based NNS on the train dataset. Finally, the authors do not discuss much how the radius $r$ should be chosen, but if I understand correctly, in the experiments of Section 5, the radius is chosen to be equal to the average cluster size, so it is defined by the clustering which may affect the measurements.

The paper also contains theoretical analysis. Theorem 1 states that clustering-NSM satisfies the properties of a clustering measure from Ben-David & Ackerman (2008). While the properties trivially follow from the definition, I find this result useful for the paper. (The only comment I have here is about the weights $w$: in the formulation of the theorem, a set of weights is assumed to be fixed, but it is not clear what it means since in the default scenario considered in the paper, the weights depend on the cluster sizes. More rigorous formulation would be helpful here.) Regarding Theorem 2, the requirements there are very strict. Not only should the dataset be clusterable into non-intersecting balls centered at the elements, but also each point should be the center of some ball in the same number of partitions. Moreover, the sizes of the balls are equal to the parameter $r$ in point-NSM. These conditions are very strict, and they are basically what is required to easily obtain the result, which makes the theorem not so informative. Also, I advise the authors to provide more details about the concentration result (formulate the inequality that is used together with all its assumptions and why they are satisfied), which would make the result easier to follow by a reader.

**Reviewer Concerns:**

The reviewers raised the following concerns:
- Possibly high computational complexity. The authors explained in their rebuttal how complexity can be reduced through approximate NNS and subsampling.
- The method predicts how hard the dataset is for ANNS, but to compute the measure, one needs to perform ANNS for the points in the dataset.
- The correlation of cluster-NSM with ANNS performance is not surprising since the measure is created to mimic the cluster-based NNS. The authors replied that this is indeed the case when there is no distribution shift for test queries.
- There were questions about the chosen external and internal measures. The authors explained their choice in the rebuttal.
- There was a concern that the point-NSM can be significantly affected if a dataset has varying density. However, as I understand, this concern holds for $k>1$, while in the paper the authors consider $k=1$.
- There were questions about the theoretical analysis, which were answered.
- The scope of the paper is limited. In particular, it does not give an answer to whether clustering-based ANNS should be used relative to other methods, which the authors say is out of the scope of this paper.
- There were questions about practical utility. The authors suggested several potential applications.

**Reviewer Scores:**

The reviewer scores are the following:
- Reviewer gKg9: 4. Concerns have been addressed in the rebuttal; the score could have improved to 6.
- Reviewer iRDX: 2. Some of the concerns of this reviewer are addressed; the score could have improved to 4.
- Reviewer h922: 4. The questions raised in the original review have been addressed. The reviewer asked one more question in the discussion, and the score remains the same after the discussion.
- Reviewer epcJ: 6. All the questions are addressed, the score is 6 after the discussion.
- Reviewer xvr7: 2. The score is not changed after the discussion.

---

### Decision · Program_Chairs · 2026-01-26

Reject